# HDX reveals the conformational dynamics of DNA sequence specific VDR co-activator interactions

Jie Zheng[1], Mi Ra Chang[1], Ryan E. Stites[2], Yong Wang[2], John B. Bruning[3], Bruce D. Pascal[1], Scott J. Novick[1], Ruben D. Garcia-Ordonez[1], Keith R. Stayrook[2], Michael J. Chalmers[2], Jeffrey A. Dodge[2] & Patrick R. Griffin[1]

The vitamin D receptor/retinoid X receptor-α heterodimer (VDRRXRα) regulates bone mineralization via transcriptional control of osteocalcin (*BGLAP*) gene and is the receptor for 1α,25-dihydroxyvitamin $D_3$ (1,25D3). However, supra-physiological levels of 1,25D3 activates the calcium-regulating gene *TRPV6* leading to hypercalcemia. An approach to attenuate this adverse effect is to develop selective VDR modulators (VDRMs) that differentially activate *BGLAP* but not *TRPV6*. Here we present structural insight for the action of a VDRM compared with agonists by employing hydrogen/deuterium exchange. Agonist binding directs crosstalk between co-receptors upon DNA binding, stabilizing the activation function 2 (AF2) surfaces of both receptors driving steroid receptor co-activator-1 (SRC1) interaction. In contrast, AF2 of VDR within VDRM:*BGLAP* bound heterodimer is more vulnerable for large stabilization upon SRC1 interaction compared with VDRM:*TRPV6* bound heterodimer. These results reveal that the combination of ligand structure and DNA sequence tailor the transcriptional activity of VDR toward specific target genes.

[1] Department of Molecular Medicine, The Scripps Research Institute, Jupiter, FL 33458, USA. [2] Lilly Research Laboratories, Eli Lilly and Company, Indianapolis, IN 46225, USA. [3] The University of Adelaide, Institute for Photonics & Advanced Sensing (IPAS), School of Biological Sciences, University of Adelaide, Adelaide SA 5005, Australia. Correspondence and requests for materials should be addressed to P.R.G. (email: pgriffin@scripps.edu)

Calcium ion metabolism and homeostasis is co-operatively governed by the intestine, kidney, and bone to ensure physiological bone mineralization, an important process of laying down calcium phosphate on bone matrix[1]. The vitamin D receptor (VDR), a member of the nuclear receptor (NR) super-family, orchestrates calcium homeostasis and bone mineralization through transcriptional control of VDR target genes in various tissues. VDR is activated by the full agonist secosteroid hormone 1,25D3[2, 3] (an active metabolite of vitamin D3) resulting in increased expression of the osteoblast hormone osteocalcin (bone gamma-carboxyglutamic acid-containing protein (*BGLAP*)), a protein essential for bone formation[4]. As such, 1,25D3 is used to treat osteoporosis, a metabolic disease that manifests clinically as deceased bone mass density. Although 1,25D3 is essential for normal osteoclast formation, hyperactivation of VDR by this secosteroid hormone results in untoward effects including hypercalcemia and hypercalciuria derived from increased calcium absorptions in the intestine[5]. The association of 1,25D3 treatment with hypercalcemia is in part related to VDR-mediated expression of the epithelial $Ca^{2+}$ channel TRPV6 (transient receptor potential vanilloid type 6). This channel mediates transepithelial calcium transport, and it is a major VDR target gene activated by 1,25D3 action in intestine[6, 7]. Therefore, safer vitamin D therapeutics for treatment of osteoporosis are in high demand that drive bone mineralization but are devoid of hypercalcemia effects. One approach has been the development of non-calcemic VDR modulators known as VDRMs. These compounds are designed to differentially activate VDR target genes. To help facilitate the design and discovery of such compounds, an understanding of the complexity of the structural and molecular mechanisms of VDRM action should be considered. This includes knowledge of the conformational changes within the VDRRXRα heterodimeric receptor complex that are induced by ligand binding and interaction with co-regulatory proteins, as well as allosteric crosstalk upon binding specific regions thought to regulate VDR target genes.

VDR is a ligand-dependent transcription factor that forms an obligate heterodimer with retinoid X receptor α (RXRα)[8–10]. Like other NRs, VDR and RXRα comprise four major functional domains. The N-terminal domain (NTD) is structurally disordered and contains the activation function 1 (AF1) motif which facilitates ligand-independent activation of the receptor. Adjacent to the NTD is the two-zinc finger-containing DNA binding domain (DBD) and a flexible hinge domain links the DBD to the ligand-binding domain (LBD). The LBD consists of three β sheets and 12 α helices, a canonical structure adopted by most NRs[11]. Helices 3, 5, and 12 of the structured LBD serve as the ligand-dependent activation function 2 (AF2) surface for complementary binding of co-regulator proteins (e.g., the p160 family of steroid receptor co-activators such as SRC1). Agonist binding in the hydrophobic pocket of LBD facilitates AF2 to adopt a conformational rearrangement that facilitates high affinity docking of co-regulatory proteins[12] possessing conserved helical nuclear receptor box (NR box) motifs (5′—LXXLL—3′) that dock into the AF2 surface[13]. These co-regulatory proteins either contain chromatin remodeling activity or are scaffolds to tether remodeling enzymes to the receptor complex. Agonists of the receptor drive interaction with co-activator proteins with histone acetyl transferase activity to relax chromatin, whereas antagonists drive interaction with co-repressor proteins with histone deacetylase activity to condense chromatin.

NRs modulate transcriptional output by binding to specific DNA sequences (response elements) within the promoter and distal regions of their target genes, and recruiting chromatin remodeling enzymes to either facilitate binding or block binding of RNA polymerase II. Upon ligand binding, VDRRXRα translocates and associates with genomic DNA. The VDRRXRα heterodimer binds with high affinity to short DNA sequence motifs known as VDR binding sequences (VBSs), which are typically hexameric direct repeats of 5′-AGGTCA-3′ separated by three base-pair spacers, a motif referred to as DR3[14–16]. Within the receptor complex, the DBDs of both VDR and RXRα and the VDR hinge form the major structural determinants involved in VBS recognition[8, 17]. Previously it was shown that DNA sequences induce unique conformations within NRs, suggesting that DNA is not only a scaffold but is a regulatory ligand for NRs[9, 18, 19]. One previous study had shown that VDRRXRα binding to DNA exerts long-range alterations of the conformational dynamics of the AF2 region impacting receptor interaction with co-receptors[9]. In that study, distinct VBS differentially impacted the conformational dynamics of the receptor complex where the DBD allosterically transmitted information to distal regions of the receptor complex involved in transcriptional activity[20]. Together, these observations suggest that DNA plays a direct role in determining the specificity of target gene activation.

Hydrogen/deuterium exchange (HDX) coupled with mass spectrometry has proven to be a robust biophysical method to probe protein conformational dynamics in the context of ligand and protein/protein interactions[9, 21–25]. Here we apply HDX to dissect the activation mechanism of VDRRXRα heterodimer DNA complex upon binding to the endogenous agonists 1,25D3, two synthetic full agonists (Cmpd1 and Cmpd2), or a novel non-calcemic VDR modulator (VDRM; Cmpd3), and to oligonucleotides representative of specific VBSs correlating to important endogenous target genes modulated by VDR. Furthermore, the impact of ligand and VBS binding on the interaction with the NR co-activator SRC1 was profiled. Combined, the results presented reveal that ligand structure and VBS sequence coordinate allosteric intra-molecular and inter- molecular communication within the VDRRXRα heterodimer to modulate AF2 dynamics impacting transcriptional activity in a DNA sequence specific, and perhaps promoter specific fashion.

## Results

**Structure of VDRMs and their *BGLAP* and *TRPV6* activation.** Compounds were tested for their hypercalcemia liabilities in *BGLAP* and *TRPV6* qPCR gene activation assays. The chemical structures of the ligands used in these studies are presented in Table 1 and these include the endogenous full agonist 1,25D3, two synthetic full agonists (Cmpd1 and Cmpd2), and one VDRM (Cmpd3), defined as a compound that displays a unique activation profile as compared with agonists. Unlike 1,25D3, Cmpds 1, 2, and 3 are non-secosteroidal tool molecules that are highly optimized toward understanding functional selectivity of bone vs. calcium effects in cells. The natural agonist 1,25D3 exhibited potent hypercalcemia effects and enhanced activation profiles in both *BGLAP* ($EC_{50}$ = 4.10 nM, 100% Max. Stim.) and *TRPV6* ($EC_{50}$ = 9.81 nM, 100% Max. Stim.) qPCR based assays. The maximum fold change over DMSO for cells treated with 1,25D3 was set to 100% maximum stimulation for both *BGLAP* and *TRPV6*, fold change for other compounds were normalized to the maximum stimulation value for 1,25D3 for each gene respectively. EC50 values are the average from the replicate runs. Cmpd1 was the most potent in the *BGLAP* activation assay ($EC_{50}$ = 1.44 nM, 94.2% Max. Stim.) and was also able to effectively activate *TRPV6* gene expression with an $EC_{50}$ of 63 nM (80.3% Max. Stim.). Cmpd2 also displayed effective activation profiles in both *BGLAP* ($EC_{50}$ = 34.92 nM, 73% Max. Stim.) and *TRPV6* ($EC_{50}$ = 395 nM, 98.2% Max. Stim.) assays. In contrast, Cmpd3 exhibited dissociated activation profiles with an $EC_{50}$ of > 10 µM (17.4% Max. Stim.) and 698 nM (75.9% Max. Stim.) in *TRPV6*

**Table 1 Ligand structures and their hypercalcemia liability activities**

| Compound | Structure | BGLAP EC50 nM (%MaxStim, n) | TRPV6 EC50 nM (%MaxStim, n) |
|---|---|---|---|
| 1,25D3 | | 4.10 (100.0, 21) | 9.81 (100.0, 49) |
| 1[a] | | 1.44 (94.2, 2) | 63.00 (80.3, 3) |
| 2[b] | | 34.92 (73.0, 6) | 395 (98.2, 3) |
| 3[a] | | 698 (75.9, 6) | >10000 (17.4, 2) |

Chemical structures of ligands—natural agonist 1,25D3 and Cmpd1–3 (Supplementary Data 1)—are shown with their respective EC50s for *BGLAP* and *TRPV6* gene activation assays, maximum stimulation values (normalized to the fold change of 1,25D3 treated cells over DMSO for both genes) and the number of independent biological replicates
[a]The secondary alcohol stereocenter is a single unknown configuration
[b]Both stereocenters are one single configuration. The configuration of the Ala stereocenter is known, the secondary alcohol configuration is not. EC50 values are the average from the replicate runs

**Table 2 Crystallographic statistics of VDR LBD and VDRM complex**

| | |
|---|---|
| Space group | P 21 21 21 |
| Cell constants $a, b, c, \alpha, \beta, \gamma$ | 44.00 Å, 52.67 Å, 105.88 Å, 90.00°, 90.00°, 90.00° |
| Resolution (Å) | 20.31-2.20 |
| % data completeness (in resolution range) | 99.9 (20.31-2.20) |
| $R_{merge}$ | 0.548 |
| $<I/\sigma(I)>$ | 4.7 (2.34-2.2 Å) |
| $R, R_{free}$ | 0.167, 0.235 |

and *BGLAP* gene activation assays, respectively. Based on these data, Cmpd3 presented a VDRM selective activation profile of *BGLAP* over *TRPV6* with significantly lower hypercalcemia liability (75.9 vs 17.4% Max. Stim.).

**Structural features of VDR LBD-VDRM interaction**. Here we present the co-crystal structure of the VDR LBD-VDRM complex at a 2.2 Å resolution (Table 2). To date, there are less than 10 structures in the PDB of non-secosteroids co-crystallized with either rat or human VDR. The VDR LBD displays the canonical α-helical sandwich similar to that observed in most NRs, and in this structure Cmpd3 assumes a curve shaped conformation accommodating the hydrophobic ligand binding pocket (LBP) (Fig. 1a). The 5-amino tetrazole amide moiety of Cmpd3 occupies the 1,25D3 A-ring region of the VDR LBP, hydrogen bonding directly with Ser237 (H3) and through water to Tyr143 (H1) and Ser278 (Loop5) (Fig. 1b and Supplementary Fig. 2a). The secondary alcohol of Cmpd3 adopts the same bifurcated positioning between His305 (Loop 6) and His397 (H11) as 1,25D3 albeit with a slightly reduced hydrogen bonding distance between both residues and the oxygen atom (2.71 Å vs. 2.81 Å), as shown in Fig. 1c. This interaction likely assists H12 forming an agonist position with its Val418 in proximity with tert-butyl group of Cmpd3. Consistent with the crystal structure data, peptides that contain these residues all displayed strong protection to solvent exchange in the corresponding HDX studies (Supplementary Fig. 2b). To compare the binding modes of Cmpd1 and Cmpd2 with Cmpd3, docking studies were performed with the VDR LBD crystal structure using ICM Pro software (Molsoft). All compounds (Cmpd1–3 and 1,25D3) occupy a similar overall orientation and binding location in within the LBD. This is shown in Fig. 1d which depicts a superimposition of Cmpd1–3 and 1,25D3 and their binding poses within the LBD.

**Differential HDX of liganded VDRRXRα heterodimer**. Greater than 85 and 90% sequence coverage was obtained for full-length VDR and RXRα, respectively (Supplementary Fig. 1a, c and e). The HDX data from all overlapping peptides were consolidated to individual amino acid values using a residue averaging

approach[26]. HDX Workbench was used to map the HDX data to the pymol structure model displayed in Supplementary Fig. 2c. Differential HDX was performed to investigate conformational perturbations within the intact VDRRXRα heterodimer induced upon binding to natural agonist, synthetic agonists and a non-calcemic VDRM. Regions within the receptor complex that exhibit a decrease in solvent exchange as compared to apo-complex are inferred to have been stabilized by the binding event. Likewise, regions within the receptor complex that exhibit increased exchange as compared with apo-complex are inferred to have been destabilized by the binding event. Binding of intact heterodimer to different ligands stabilized regions of the receptor corresponding to the consensus ligand binding sites of VDR and dimer interface within RXRα (RXRα: aa347–353 and aa419–430) (Fig. 2a–d and Supplementary Fig. 1b, d, column (i), (ii), (iii), and (iv)). VDR residue regions—aa134–150 (H1), aa225–233 (H3), aa273–279 (H5), aa310–316 (H7), aa390–403 (H11) and aa411–419 (H12)—exhibited significantly reduced HDX kinetics upon binding to natural ligand 1,25D3 (Fig. 2a and Supplementary Fig. 1b, d, column (i)). Consistent with the structure of 1,25D3 bound VDR LBD (PDB: 1DB1), residues within these α-helixes—Tyr143 (H1), Ser237 (H3), Arg274 (H5), Ser278 (H5), Trp286 (S1) and His305 (loop between H6 and H7)–form a hydrophobic ligand-binding pocket (LBP) within the VDR LBD fold for accommodation of 1,25D3[10, 12, 15]. H11 and H12 (His397, Val418 and Phe422) whose positions are critical to AF2 activation adopt an agonist position and make contacts with the methyl group of 1,25D3 via van der Waals's force in the reported X-ray crystal structure[12]. The ligand-dependency of H12 position is known to be critical for forming a co-activator binding surface that allows recruitment of co-activator proteins[27]. Binding of Cmpd3 to heterodimer resulted in lower threshold of protection against deuterium exchange at residue regions—aa234–244 (H3), aa309–316 (H7), and aa390–403 (H11) (Fig. 2d and Supplementary Fig. 1b, d (column iv)). Despite being a weaker binder, interaction with this non-calcemic ligand afforded a similar extent of protection at VDR H12 compared to that of 1,25D3. Furthermore, binding of synthetic agonist Cmpd1 or Cmpd2 presented even higher magnitude of protection against solvent exchange at aforementioned regions compared to 1,25D3, suggesting that synthetic agonists associate with stronger hydrogen bonding networks within VDR LBP (Fig. 2b, c and Supplementary Fig. 1b, d column (ii) and column (iii)). Furthermore, binding of Cmpd3 resulted in protection to solvent exchange in the VDR hinge domain/C-terminal Extension (CTE) (aa93–99 and aa99–108) and RXRα residues 271–293 (H3), and 439–450 (H11) that were not present for natural or synthetic agonists, suggesting a distinct ligand binding property for this selective modulator (Fig. 2d and Supplementary Fig. 1b, d column (iv)). Remote from VDR LBP, the hinge domain could be possibly stabilized either through direct binding with ligand itself or allosteric intra-molecular

interactions with VDR LBD. Interestingly, DNA binding could also disrupt the hydrogen bonding activities of VDR hinge domain[9, 28]. It is hence possible that ligand binding could influence DNA binding activity by directly impacting the conformation of DNA-interacting hinge domain. Furthermore, binding of all VDR ligands led to allosteric destabilization or increased solvent exchange at residues aa82−89 derived from VDR DBD, indicating that ligand binding could induce allosteric intramolecular crosstalk and potentially modulate DBD function (Fig. 2a−d and Supplementary Fig. 1b, d column (i), (ii), (iii), and (iv)).

**LBD and hinge are functionally integrated with the DBD.** Next, we introduced oligonucleotides representative of two hypothetical yet distinct VBSs to investigate how DNA binding events affect conformational dynamics of VDRRXRα heterodimer when it is bound with the natural agonist 1,25D3. One synthetic oligo (DR3) represents an idealized direct repeat DR3 VBS (5′-CGTAGGTCAATCAGGTCACGTCGT-3′) containing two consensus half-sites 5′-AGGTCA-3′ separated by three base pairs, whereas the other oligo (DR3 half-site) contains only one consensus half-site (5′-CTAGCTCCCGAGGTCAGCGACGG

CGCAGG-3′). Interaction of the VDRRXRα heterodimer with either of the two hypothetical VBSs resulted in protection to solvent exchange in DBD regions of both receptor and co-receptor (Fig. 3a, b and Supplementary Fig. 1b, d column (v) and (vi)). Specifically, VDR residues aa17−25 containing zinc-coordinating residue Cys24 exhibited a twofold protection with DR3 half-site compared to DR3 VBS, suggesting perhaps that interaction with DR3 half-site VBS requires an alternative conformation conferred at the zinc finger (Supplementary Fig. 1b column (v) and column (vi)). Interestingly, VDR hinge/CTE domain afforded a higher magnitude of protection to solvent exchange upon binding of DR3 when compared to binding to DR3 half-site VBS (Supplementary Fig. 1b column (v) and column (vi)). VDR hinge/CTE domain, which contains positively charged residues Arg102, Lys103, and Arg103, might form stronger electrostatic interactions with negatively charged phosphate backbone of the DR3 oligo compared with that of DR3 half-site oligo. CTE-DNA interactions have also been observed in PPARγRXRα and Rev-Erb structures with their CTEs inserted into the minor groove of DNA[27, 29]. Unlike VDR, RXRα DBD (aa137−158 and aa169−178) presented analogous HDX profiles upon binding to DR3 half-site and DR3 VBSs (Supplementary

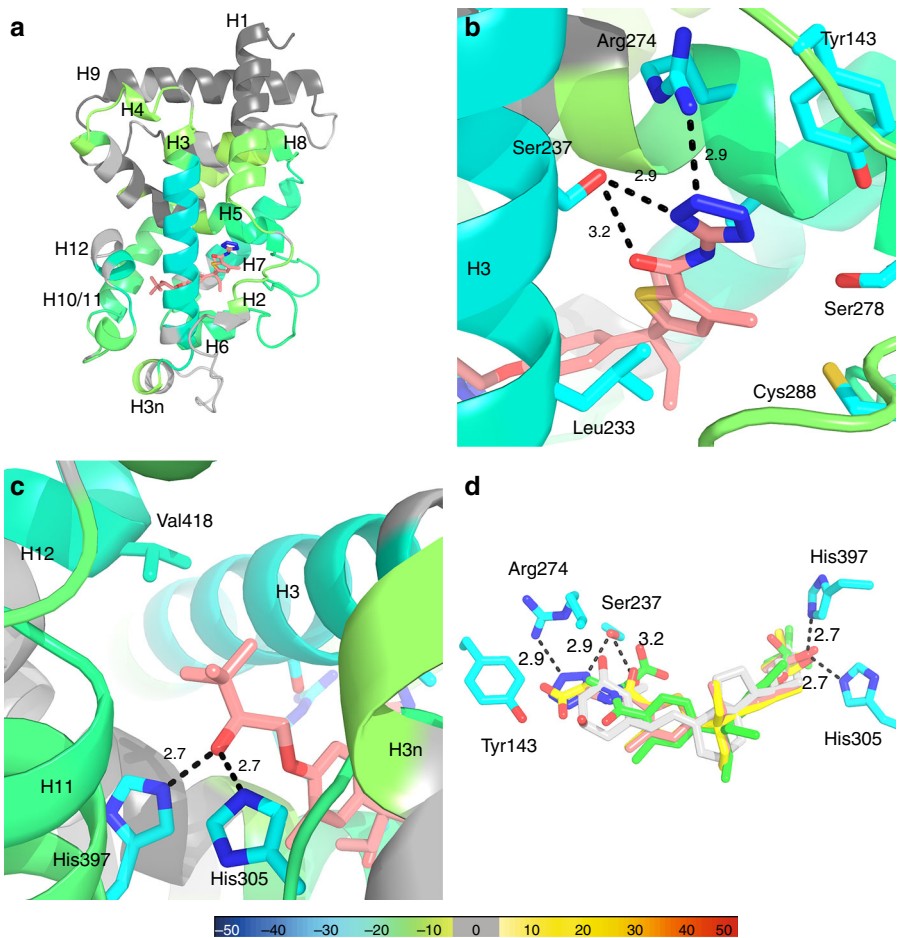

**Fig. 1** Structural features of VDR LBD-VDRM interaction. **a** Ribbon diagram of the VDRM bound LBD crystal structure of VDR. The ribbon diagram is colored by HDX stabilization/destabilization. Percentages of deuterium differences are color-coded according to the smooth color gradient key at the bottom of Fig. 1 (Supplementary Fig. 2b). **b** Crystal structure of the VDR LBD region encompassing the pocket near H3/beta-sheet in complex with Cmpd3 (pink). Residues making interactions with Cmpd3 are shown in sticks. Dashes represent hydrogen bonds. **c** Crystal structure of the VDR LBD region encompassing the pocket near H12/AF2 region in complex with Cmpd3 (pink). Residues making interactions with Cmpd3 are shown in sticks. Dashes represent hydrogen bonds. **d** Superimposition of Cmpds 1–3 (X-ray crystal structure of Cmpd 3 and docked structures for Cmpds 1 and 2). The VDR LBD residues are shown in sticks. Cmpds's 1, 2, 3, and 1,25D3 (PDB:1DB1) are colored in green, yellow, pink, and white, respectively. The oxygen and nitrogen atoms are shown as red and blue, respectively

Fig. 1d column (v) and column (vi)). Such stabilization of the heterodimer could be either derived from protein-DNA interaction or heterodimerization between the co-receptor's DBDs[27].

The most striking differences in HDX kinetics upon binding the different VBSs was within the VDR LBD. The binding of DR3 half-site VBS resulted in substantial protection to solvent exchange in regions constituting the AF2 surface (aa300−308 (H5), aa391−403 (H11), and aa411−419 (H12)), as well as other regions within the VDR LBD; e.g., aa136−150 (H1), aa206−218 (loop) and aa245−259 (H4) (Fig. 3b and Supplementary Fig. 1b, d column (vi)). Specifically, VDR residues aa109−133 residing on the C-terminus of hinge/CTE and N-terminus of helix 1 (H1) underwent partial cooperative unfolding event (EX1 kinetics as revealed by detection of two distinct deuterated ion distributions for the same peptide[30]) upon heterodimer binding to DR3 half-site VBS (Supplementary Fig. 2d). For EX1 kinetics the refolding rate of the corresponding protein region is slower than the solvent exchange rate of amide hydrogens in the unfolded region, thus these amide hydrogens exchange simultaneously. This situation gives rise to a distinct MS signature, specifically bimodal mass distributions, with the lower mass envelop corresponding to molecules that have not yet exchanged (not yet unfolded) and the higher mass envelop corresponding to molecules that have undergone solvent exchange (molecules that have unfolded). Under native state conditions, proteins exhibit EX2 kinetics wherein only a single MS envelop is observed over time (the refolding rate is faster than the rate of solvent exchange). However, the occurrence of EX1 cooperative unfolding behavior in native state proteins have been shown to provide important clues to intermediate conformational states of proteins[22, 31–33]. The occurrence of EX1 behavior indicates that the hinge/H1 region of VDR is structurally heterogeneous in solution when 1,25D3-heterodimer is bound to DR3 half-site VBS, perhaps a mixture of molecules in either an open or closed conformation. These two distinct conformations might differentially orient the LBD with respect to the DBD. Unlike that observed upon binding

DR3 half-site VBS, little conformational change was observed in VDR LBD when bound to DR3 (Fig. 3a, b and Supplementary Fig. 1b, d column (v) and (vi)), and the hinge/H1 region of VDR adopted a homogeneous conformation as only EX2 kinetics were observed (Supplementary Fig. 2d). In the previously reported cryo-EM structure of VDRRXRα bound to DR3 VBS, Pro122, which resides on the VDR hinge, was described as the "kink residue" between the C-terminal hinge helix and helix H1 of the LBD playing an important role in dictating the orientation of the LBD, resulting in an open LBD dimer architecture facing away from DBD[8]. It is thus reasonable to propose that the DBD/hinge domains of VDR confer target VBS sequence specificity and selectively transmit allosteric signals to alter the dynamics of the LBD. In contrast, the LBD of RXRα afforded similar HDX kinetics upon binding to either of the two VBSs (Fig. 3a, b and Supplementary Fig. 1d column (v) and (vi)). The RXRα dimer interface, aa420−430 (H10), showed reduced HDX, suggesting that ligand and DNA binding sequentially stabilize the VDRRXRα heterodimer, and that VDR heterodimerization with RXRα likely enhances binding affinity for DNA[8, 9, 34]. Importantly, as shown in Supplementary Fig. 2e, the ligand-free heterodimer binds to these two different VBSs with similar affinity as determined using an electrophoretic mobility shift assay (EMSA). This observation further supports that the sequence of the VBS, and not its affinity for heterodimer, correlates with alterations of AF2 dynamics within VDR that impact intra-molecular signaling and target gene transactivation.

To further investigate whether compounds with distinct chemical structures induce unique conformations within the heterodimer upon DNA binding, parallel HDX experiments were conducted in the presence of synthetic agonists to probe conformational dynamics of VDRRXRα heterodimer with and without DR3 and DR3 half-site VBSs, respectively. The synthetic agonists, Cmpd1 and Cmpd2, induced a greater magnitude of protection to solvent exchange across the DBD and hinge domain of VDR upon binding to DR3 VBS resulting in larger ΔΔHDX

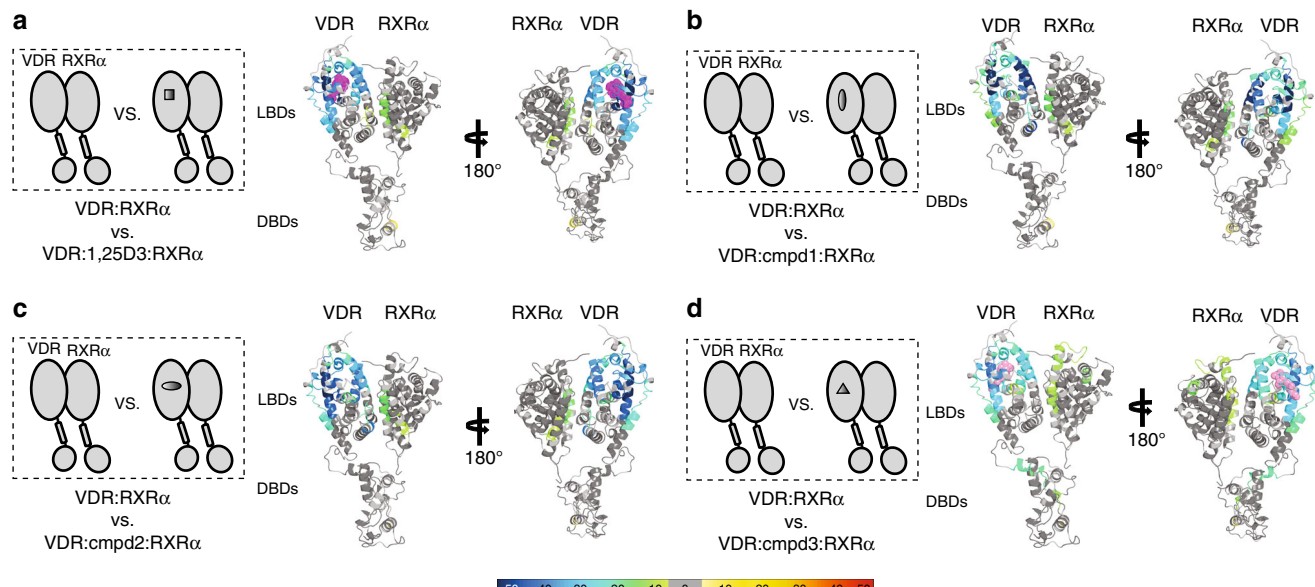

**Fig. 2** HDX characterization of ligand binding effects within intact VDRRXRα heterodimer. Schematic representations illustrate differential experiments of VDRRXRα heterodimer verse VDRRXRα: ligand complex (on the left). Differential consolidation HDX data are mapped onto the full-length VDRRXR heterodimer structure model in ribbon (on the right), as shown by representation of altered conformational dynamics of co-receptors upon binding to **a** natural ligand 1,25D3 (Supplementary Fig. 1b, d, column (i)), **b** Cmpd1 (Supplementary Fig. 1b, d, column (ii)), **c** Cmpd2 (Supplementary Fig. 1b, d, column (iii)), and **d** non-calcemic ligand Cmpd3 (Supplementary Fig. 1b, d, column (iv)). Percentages of deuterium differences are color-coded according to Fig. 1. Dark gray, no statistically significant changes between compared conditions; light gray, regions that have no sequence coverage and include proline residue that has no amide hydrogen exchange activity; purple, 1,25D3 ligand; pink, Cmpd3

between the two VBS binding datasets ($\Delta\Delta HDX = \Delta HDX_{heterodimer:agonist \pm DR3} - \Delta HDX_{heterodimer:agonist \pm DR3\ half-site}$). For instance, Cmpd1 and Cmpd2 induced a 21 and 19% $\Delta\Delta HDX$ ($p < 0.001$) in the VDR hinge region (aa97−108) whereas only 4% $\Delta\Delta HDX$ was observed when bound with 1,25D3 (Fig. 3c−f and Supplementary Fig. 1b, column (vii), (viii), (ix), and (x)). However, the most striking difference was observed within a region of the VDR LBD that possessed insensitive to the VBS sequence (Fig. 3c−f). In this region, binding of synthetic agonists and DNA resulted in protection to solvent exchange in LBD regions spanning H1, loop, H3, H5, H9, H11, and H12 (Fig. 3c−f and Supplementary Fig. 1b column (vii), (viii), (ix), and (x)). This was a dramatic contrast to that observed for 1,25D3-bound heterodimer upon binding DR3 and DR3 half-site VBSs, which each induced unique HDX signatures within the VDR LBD. In addition, the VDR hinge/H1 peptide (aa109−133) underwent partial unfolding (as revealed by observing EX1 kinetics) when heterodimer was bound to synthetic agonists and DNA (Supplementary Fig. 2d). It is possible that the presence of synthetic agonists and DNA drive structural rearrangement in VDR hinge/H1 to affect hydrogen bonding networks within the LBD. Such a scenario may suggest that transduction of signal from the receptors' DBD to its LBD could be regulated not only by specific VBS sequence but also by the specific ligand in the LBP. Therefore, at least for VDR, the DBD appears to be

functionally integrated with LBD in response to the specific sequence of a VBS within a target gene promoter, as well as the exact chemical structure of the ligand.

Interaction with synthetic agonists and VBSs drove unique HDX perturbation patterns in regions involved in RXRα hinge domain as well as its AF2 surface. On the RXRα side of the heterodimer, reduced solvent exchange was observed within the DBD (aa137−158, aa169−178), the dimer interface (aa419−430), hinge-H1 (aa220−237) region, and the loop region between H8 and H9 domain (aa376−390), which is oriented within the AF2 region (Fig. 3c−f and Supplementary Fig. 1d column (vii), (viii), (ix), and (x)). Previously it has been shown that the hinge domain of NRs plays a critical role in maintaining the activity of NRs[28]. The presence of Cmpd2 and DNA afforded additional protection to solvent exchange in H12 AF2 helix (aa452−462) of RXRα (Fig. 3e, f and Supplementary Fig. 1d column (ix) and (x)). To further examine whether these distinct HDX signatures are specific to heterodimer but not the RXRα co-receptor, we performed HDX studies of RXRα alone in the presence and absence of ligand and DNA. The binding of Cmpd1 to RXRα alone preserved apo-like dynamics suggesting that there are no detectable conformational changes of RXRα upon ligand binding (Supplementary Fig. 1d column (xxvi)). Sequential interaction with DR3 VBS resulted in protection to solvent exchange only in RXRα DBD (Supplementary Fig. 1d column (xxvii)). The absence

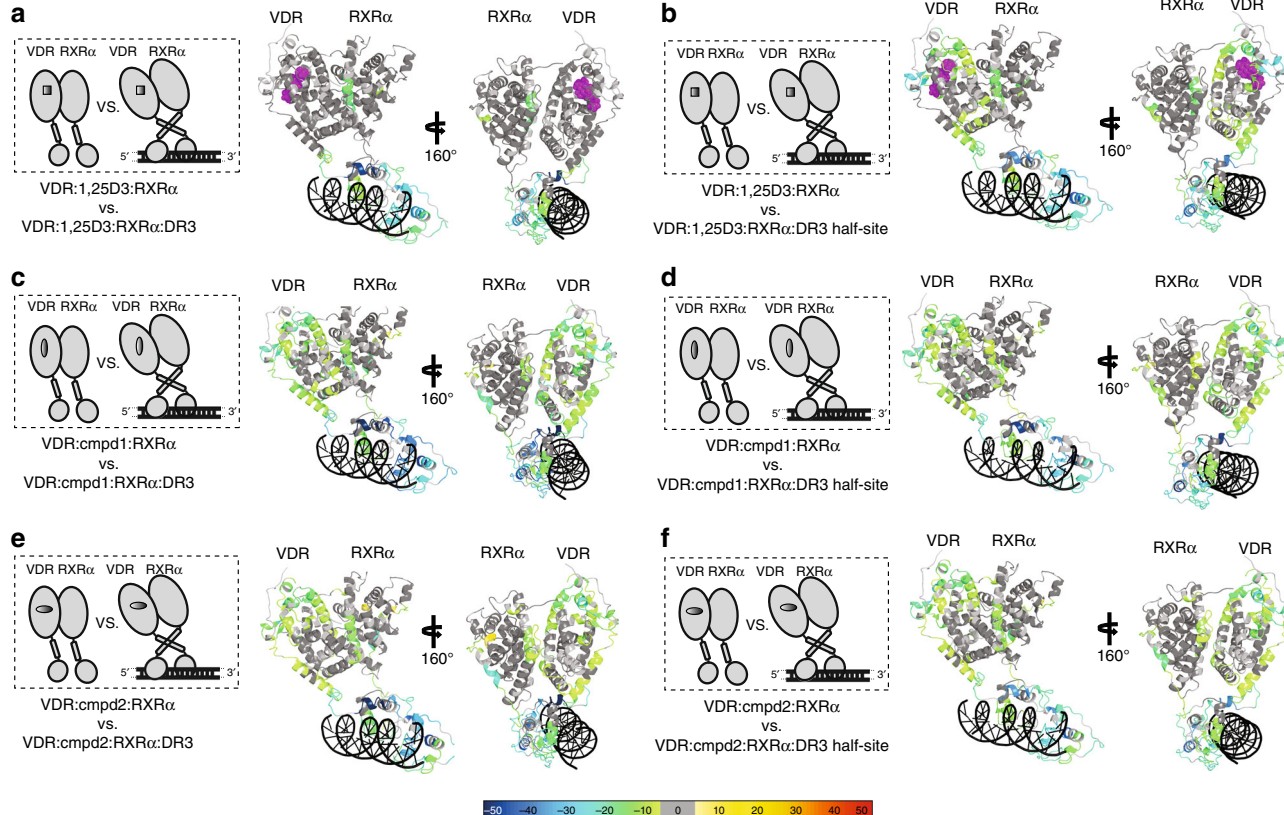

**Fig. 3** Comparisons of DR3 and DR3 half-site binding effects on natural agonist- and synthetic-agonist- bound heterodimer complexes. Schematic representations illustrate differential experiments of VDRRXRα: ligand complex verse VDRRXRα: ligand: DNA complex (on the left). Consolidated differential HDX data are mapped onto the VDRRXRα heterodimer structure model in ribbon when the complex is bound to different ligands and DNA VBS (on the right). Conformational changes of 1,25D3-bound heterodimer upon binding of **a** DR3 (Supplementary Fig. 1b, d, column (v)) and **b** DR3 half-site (Supplementary Fig. 1b, d, column (vi)). Conformational changes of Cmpd1 bound heterodimer upon binding of **c** DR3 (Supplementary Fig. 1b, d, column (vii)) and **d** DR3 half-site (Supplementary Fig. 1b, d, column (viii)). Conformational changes of Cmpd2 bound heterodimer upon binding of **e** DR3 (Supplementary Fig. 1b, d, column (ix)) and **f** DR3 half-site (Supplementary Fig. 1b, d, column (x)). Percentages of deuterium differences are coded as Fig. 1. Dark gray, no statistically significant changes between compared conditions; light gray, regions that have no sequence coverage and include proline residue that has no amide hydrogen exchange activity; purple, 1,25D3 ligand; black, DNA VBSs

of VDR abrogated those unique HDX perturbations in RXRα. Upon heterodimerization with VDR, the dynamic nature of RXRα in the synthetic agonist bound state differs from that of 1,25D3 upon interacting with DNA. Therefore, these results suggest that ligand and DNA could both coordinate both intra-molecular and inter-molecular allosteric communications within a NR heterodimer complex.

**Agonists stabilize AF2 of both receptors upon SRC1 binding**. The AF2 region of NRs serves as the binding surface for recruitment of transcriptional co-regulatory proteins that contain an α-helical nuclear receptor binding motif referred to as an NR box[9, 35]. To test the hypothesis that the alterations in the stability of the heterodimer driven by ligand and DNA binding could modulate co-activator interaction, we sequentially probed the conformational ensembles of heterodimer complex in the presence and absence of the receptor interaction domain (RID) of the p160 NR co-activator steroid receptor co-activator 1 (SRC1 RID). The RID of SRC1 contains three LXXLL NR box motifs. In the presence of 1,25D3, interaction with SRC1 RID resulted in ∼ 21 and ∼ 16% reduction in solvent exchange ($p < 0.001$) within H12 (aa411–419) of VDR when the 1,25D3-bound heterodimer was in the presence of DR3 and DR3 half-site VBSs, respectively (Fig. 4a, b and Supplementary Fig. 1b column (xi) and (xii)). In contrast to that observed on the VDR side of the 1,25D3-bound heterodimer, the RXRα AF2 surface was insensitive to the presence of the SRC1 RID, further suggesting ligand dependency for SRC1 recruitment to the heterodimer is associated with exclusively the VDR AF2 surface, and upon binding forms a compact protein complex when bound to either DR3 and DR3 half-site VBSs. These HDX data are consistent with observations made using SAXS[36]. In this study, it was concluded that the NR heterodimers binds only one molecule of SRC1 RID and does so through the RXRα partner[36]. Binding of SRC1 RID to either Cmpd1-heterodimer-DR3 or Cmpd1-heterodimer-DR3 half-site complex resulted in nearly indistinguishable HDX patterns within H12 (aa412–419) of VDR with ∼ 11% decrease in solvent exchange (Fig. 4c, d and Supplementary Fig. 1b, column (xiii) and (xiv)). Similar effects on HDX kinetics were observed for both Cmpd2-heterodimer-VBS complexes where interaction with SRC1 RID resulted in decreased exchange of ∼ 20% for H12, ∼ 9% for H3, and ∼ 6% for H1 (Fig. 4e, f and Supplementary Fig. 1b, column (xv) and (xvi)). In contrast to that observed with 1,25D3, in addition to stabilization of VDR AF2, binding of SRC1 RID stabilized regions of RXRα AF2 (aa211–279 (H3), aa438–449 (H11–12), and aa439–450 (H11–12) (Fig. 4c–f and Supplementary Fig. 1d column (xiii), (xiv), (xv), and (xvi)). Previously, it has been shown that residues Phe277 (H3), Phe437 (H11), and Phe450 (H12) are important for formation of a stable AF2 surface facilitating co-activator binding to RXRα[15]. In the absence of VDR, RXRα alone was not capable of binding to SRC1 RID (Supplementary Fig. 1d column (xxviii)). Combined, these data suggest that synthetic agonist and DNA binding to the VDRRXRα heterodimer results in distinct VDR and RXRα orientations, which facilitate co-activator interaction and that synthetic agonists, but not 1,25D3, modulate inter-allosteric communications within VDRRXRα heterodimer to fine tune transcriptional activity.

**Cmpd3 vs. 1,25D3 liganded VDRRXRα on BGLAP and TRPV6 VBSs**. HDX was employed to probe the structural dynamics and activation mechanism of 1,25D3 and the non-calcemic modulator Cmpd3 liganded heterodimer bound to oligonucleotides representative of VBSs linked to the VDR target genes *BGLAP* and *TRPV6*. The sequences of the oligonucleotides used were derived from VBSs reported in *BGLAP* and *TRPV6*, each containing two direct repeats separated by a three base-pair gap, a motif of a family of closely related VBSs[36, 37]. For BGLAP two distinct oligos were used. The first was representative of a VBS located at 457 b upstream from the transcription start site of *BGLAP* from rattus (5′-CTAGGTGAATGAGGACAT-3′), a VBS used in the previously reported SAXS study. A second oligo was representative of a VBS located at 510 b upstream of start site of human *BGLAP* gene (5′-GGTGACTCACCGGG TGAACGGGGGCA-3′)[38]. For *TRPV6*, an oligo representative of a VBS located 4.3 kb upstream from the transcription start site of human *TRPV6* gene (5′-CAAGGGGTAGTGAGGTCA AAAGCA-3′) was used. Binding of 1,25D3 liganded heterodimer to *TRPV6* VBS resulted in higher protection to solvent exchange in both the DBD and hinge domains of VDR as compared to that observed upon interaction with the *BGLAP* VBS, indicating that the *TRPV6* VBS makes more interactions with these domains than does the *BGLAP* VBS (Fig. 5a, b and Supplementary Fig. 1b, column (xvii) and (xviii)). VDR LBD exhibited only subtle perturbations in HDX behavior upon binding to the *BGLAP* VBS (Fig. 5a, i and Supplementary Fig. 1b, column (xvii)). This observation is consistent with previous SAXS results that suggest the heterodimer complex forms an elongated conformation when bound to *BGLAP* VBS[36]. We then analyzed the protein complex bound to the human *BGLAP* VBS and obtained similar results to that with the rat VBS (Supplementary Fig. 1b, column (xvii*)). Similarly, binding of 1,25D3 liganded heterodimer to *TRPV6* VBS displayed minor alterations in the dynamics of the VDR LBD (Fig. 5b, i and Supplementary Fig. 1b column (xviii)). Additionally, RXRα exhibited reduced deuterium exchange only in its DBD and dimer interface upon binding to *BGLAP* and *TRPV6* VBSs (Fig. 5a, b and Supplementary Fig. 1b, column (xvii) and (xviii)). Combined, these results further support that in the presence of the natural ligand 1,25D3, the heterodimer forms an open conformation upon binding to either *BGLAP* or *TRPV6* VBSs regardless of their specific nucleotide sequence.

HDX analysis of ligand binding of VDRM Cmpd3 already revealed that it processes lower potency to perturb VDR LBP dynamics compared to that of 1,25D3 (Fig. 2d and Supplementary Fig. 1b column (iv)). The presence of Cmpd3 and *TRPV6* VBS resulted in further stabilization of the DBD and hinge domains of VDR as compared to the presence of the *BGLAP* VBS, as shown by higher extent of protection to solvent exchange within these regions (Fig. 5c, d and Supplementary Fig. 1b column (xix) and (xx)). Unlike the effects of 1,25D3, the binding of *BGLAP* and *TRPV6* VBSs to the VDRM-bound heterodimer induced different degrees of protection to solvent exchange within the VDR LBD (Fig. 5c, d and Supplementary Fig. 1b column (xix) and (xx)). Nearly identical HDX profiles were observed for the protein-ligand complex bound to either the rat *BGLAP* VBS or the human *BGLAP* VBS (Supplementary Fig. 1b, column (xix*)). A further comparison of the HDX data revealed that a larger surface area of VDR LBD demonstrated increased protection to solvent exchange in the presence of the *TRPV6* VBS when compared with the presence of the *BGLAP* VBS (Fig. 5c, d and Supplementary Fig. 1b column (xix) and (xx)). Furthermore, binding of *TRPV6* VBS to the heterodimer altered the dynamics of the VDR AF2 H12 (further reduction in deuterium exchange by ∼ 13%) that was not observed upon binding the *BGLAP* VBS (Fig. 5c, d, and j and Supplementary Fig. 1b column (xix) and (xx)). Additionally, RXRα exhibited very similar protection to exchange in its DBD and dimer interface upon binding to either *BGLAP* or *TRPV6* VBSs (Fig. 5c, d and Supplementary Fig. 1d column (xix) and (xx)). Also, no statistically significant inter-molecular communication was observed between the co-receptors. These data demonstrate that

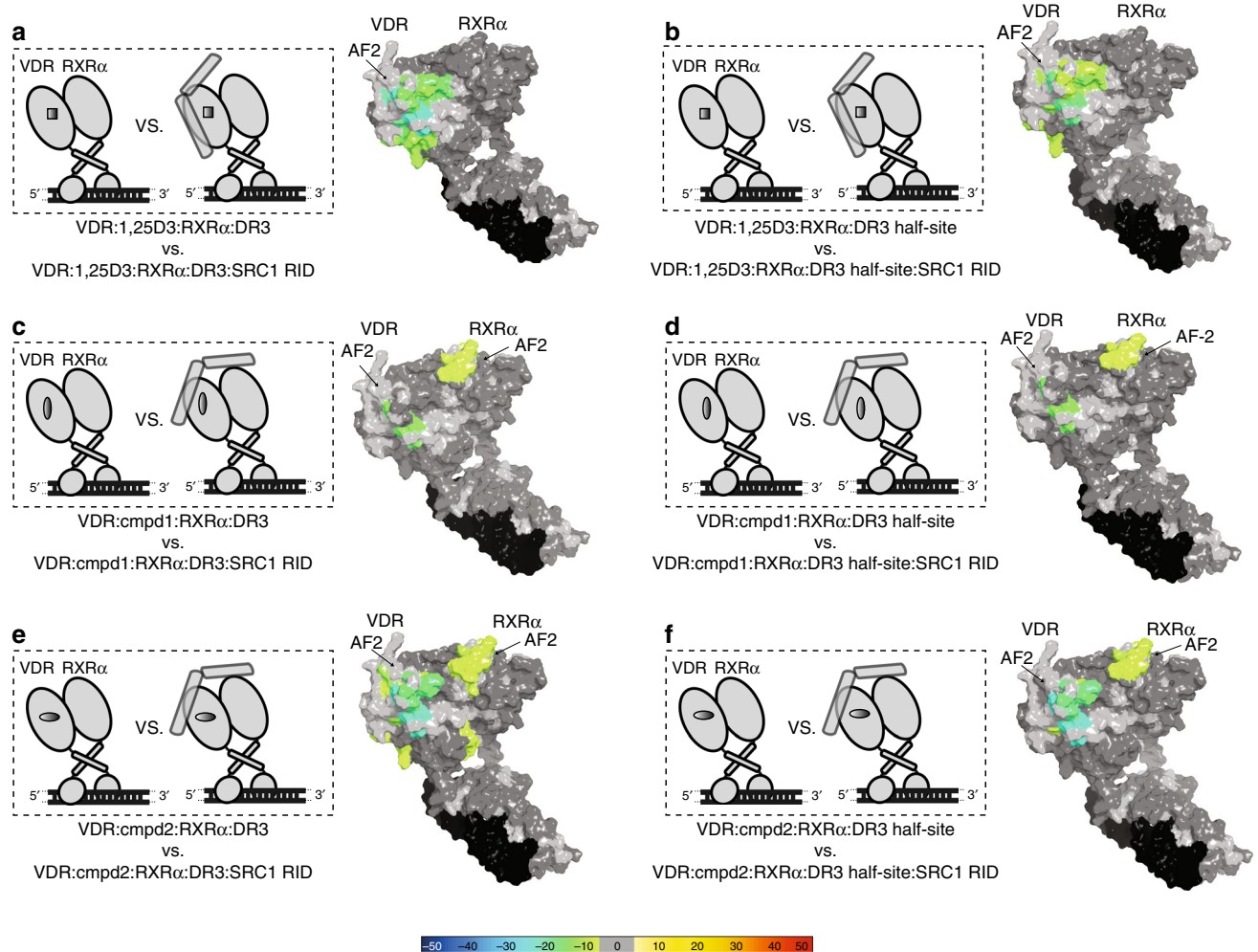

**Fig. 4** Ligand dependency of SRC1 RID binding to the VDRRXRα heterodimer AF2 surface. Schematic representations illustrate differential experiments of VDRRXRα: ligand: DNA complex verse VDRRXRα: ligand: DNA: SRC1 RID complex (on the left). Differential HDX data are mapped onto the surface of VDRRXR heterodimer structure model in the presence and absence of SRC1 RID (on the right). **a** Conformational changes of VDRRXRα: 1,25D3: DR3 complex upon SRC1 RID binding (Supplementary Fig. 1b, d, column (xi)). **b** Conformational changes of VDRRXRα: 1,25D3: DR3 half-site complex upon SRC1 RID binding (Supplementary Fig. 1b, d, column (xii)). **c** Conformational changes of VDRRXRα: Cmpd1: DR3 complex upon SRC1 RID binding (Supplementary Fig. 1b, d, column (xiii)). **d** Conformational changes of VDRRXRα: Cmpd1: DR3 half-site complex upon SRC1 RID binding (Supplementary Fig. 1b, d, column (xiv)). **e** Conformational changes of VDRRXRα: Cmpd2: DR3 complex upon SRC1 RID binding (Supplementary Fig. 1b, d, column (xv)). **f** Conformational changes of VDRRXRα: Cmpd2: DR3 half-site complex upon SRC1 RID binding (Supplementary Fig. 1b, d, column (xvi)). Percentages of deuterium differences are coded as Fig. 1. Dark gray, no statistically significant changes between compared conditions; light gray, regions that have no sequence coverage and include proline residue that has no amide hydrogen exchange activity; black, DNA VBSs

Cmpd3, a non-calcemic modulator, perturbs the dynamics of the heterodimer in an alternative mechanism from the natural ligand 1,25D3 in the presence of the *BGLAP* and *TRPV6* VBSs.

To test whether these observed differences in structural dynamics could influence receptors binding to SRC1, we performed sequential HDX studies to probe the heterodimer in the presence of RID of the co-activator. Addition of the SRC1 RID to the 1,25D3-heterodimer-*BGLAP* complex resulted in increased stabilization of the VDR AF2 surface, with minimal differences as compared to that observed for the SRC1 RID:1,25D3-heterodimer-*TRPV6* complex. HDX analysis of both complexes detects similar decrease in solvent exchange within H12 residues aa411-419 of VDR (Fig. 5e, f, and i and Supplementary Fig. 1b column (xxi) and (xxii)). These data suggest that the open H12 conformation of the 1,25D3-heterodimer DNA complex results in similar SRC1 interaction. In contrast, HDX analysis of complexes of Cmpd3-heterodimer-

*BGLAP* VBS and Cmpd3-heterodimer-*TRPV6* VBS revealed differential interaction of the SRC1 RID with the VDR AF2 surface. There was nearly a twofold higher magnitude of protection to solvent exchange in the former complex (~ 20%) when compared to the latter (~ 9%) (Fig. 5g, h, and j and Supplementary Fig. 1b column (xxiii) and (xxiv)). This difference in stabilization of VDR AF2 would suggest a difference in affinity for SRC1 binding further indicating that the VDR AF2 surface within the Cmpd3-heterodimer-*BGLAP* VBS complex facilitates SRC1 binding more than when the receptor is bound to *TRPV6* VBS. The apparent coordinated actions of Cmpd3 and *TRPV6* VBS may selectively induce conformational changes in AF2 that influence the receptors' ability to recruit transcriptional machinery. These comparative HDX studies of the VDRRXRα heterodimer in the presence of two functionally distinct ligands, 1,25D3 and the non-calcemic Cmp3, provide structural insights into how these compounds might be differentially controlling VDR target gene expression and suggest that the non-calcemic ligand Cmpd3

perturbs heterodimer dynamics in an alternative mechanism from 1,25D3 at *BGLAP* and *TRPV6* VBSs.

## Discussion

VDR controls expression of a gene program regulating calcium hemostasis. Despite its beneficial effects for treatment of osteoporosis, usage of the VDR agonist 1,25D3 is limited by dose-dependent hypercalcemia[2, 39, 40]. To improve the therapeutic index of VDR ligands, efforts have focused on the development of 1,25D3 analogs and synthetic non-secosteroid ligands that exhibit a lower threshold for hypercalcemia. Such dissociated VDR

ligands, called VDRMs, are positive on *BGLAP* expression and neutral on *TRPV6* expression. Understanding the structural mechanism that affords this dissociated profile would greatly facilitate drug development. To address this, we performed a detailed comparative biophysical study with 1,25D3, two agonists Cmpds1–2, and a non-calcemic VDRM Cmpd3. These molecules have distinct chemical structures and are distinguished based on their hypercalcemia liability profile in a *TRPV6* gene activation assay. To understand the conformational dynamics induced by these ligands upon binding VDR, differential HDX studies were performed on the VDRRXRα heterodimer in the presence and

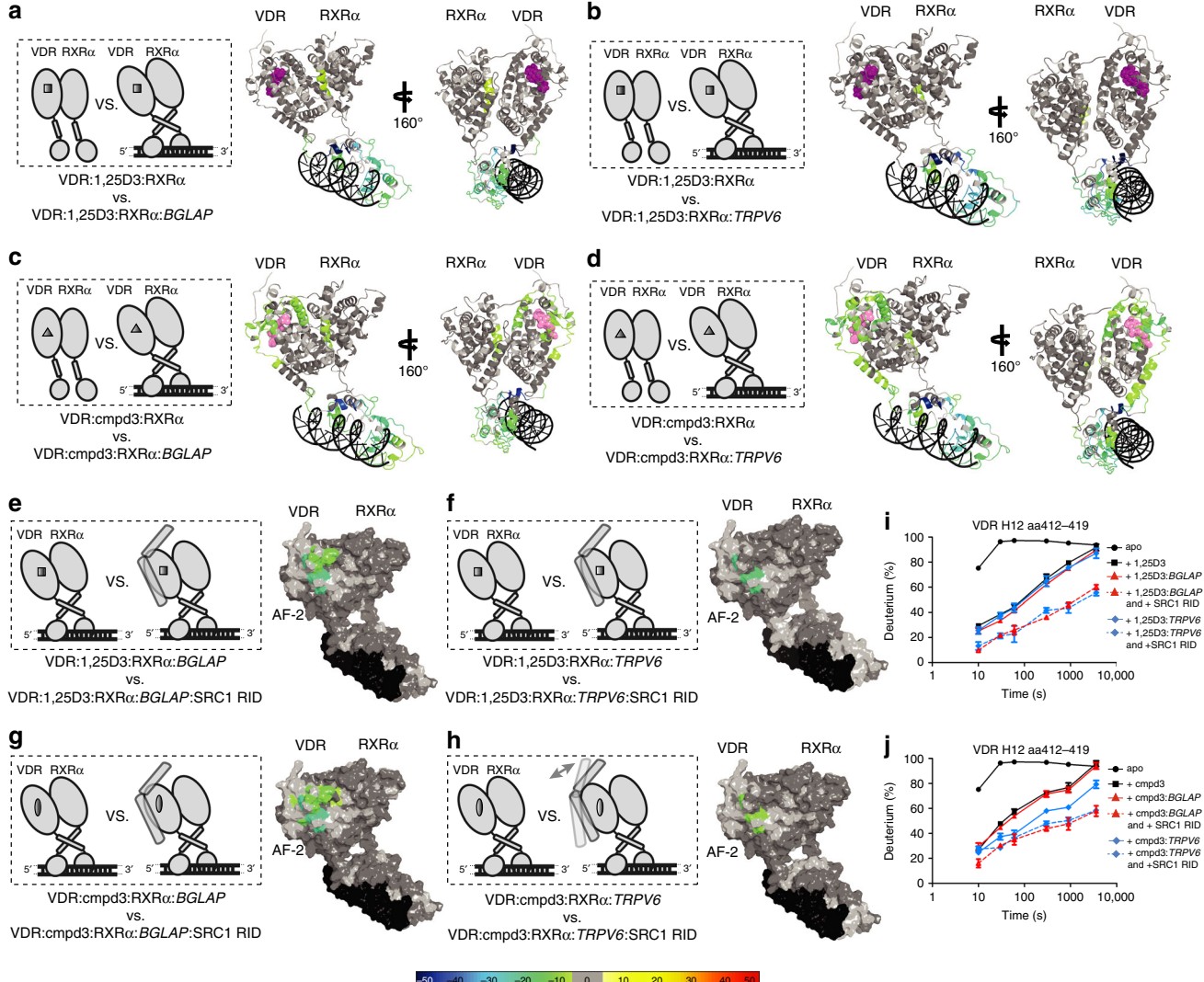

**Fig. 5** Activations of VDRRXRα heterodimer with cognate VBS *BGLAP* and *TRPV6* by selective modulator Cmpd3 verse natural ligand 1,25D3. Schematic representations illustrate differential experiments of ligand bound VDRRXRα heterodimer with respect to DNA and SRC1 RID binding (on the left). Differential consolidation HDX data are mapped onto the VDRRXR heterodimer structure model when the complex is bound to different ligands and DNA (shown in ribbon **a**–**d**), and SRC1 RID (shown in surface **e**–**h**) (on the right). Conformational changes of 1,25D3-bound heterodimer upon binding of **a** *BGLAP* VBS (Supplementary Fig. 1b, d, column (xvii)) and **b** *TRPV6* VBS (Supplementary Fig. 1b, d, column (xviii)). Conformational changes of selective modulator Cmpd3 bound heterodimer upon binding of **c** *BGLAP* VBS (Supplementary Fig. 1b, d, column (xix)) and **d** *TRPV6* VBS (Supplementary Fig. 1b, d, column (xx)). **e** Conformational changes of VDRRXRα: 1,25D3: *BGLAP* complex upon SRC1 RID binding (Supplementary Fig. 1b, d, column (xxi)). **f** Conformational changes of VDRRXRα: 1,25D3: *TRPV6* complex upon SRC1 RID binding (Supplementary Fig. 1b, d, column (xxii)). **g** Conformational changes of VDRRXRα: Cmpd3: *BGLAP* complex upon SRC1 RID binding (Supplementary Fig. 1b, d, column (xxiii)). **h** Conformational changes of VDRRXRα: Cmpd3: *TRPV6* complex upon SRC1 RID binding (Supplementary Fig. 1b, d, column (xxiv)). Percentages of deuterium differences are coded as Fig. 1. Dark gray, no statistically significant changes between compared conditions; light gray, regions that have no sequence coverage and include proline residue that has no amide hydrogen exchange activity; black, DNA VBSs; purple, 1,25D3 ligand; pink, Cmpd3. **i** Differential deuterium uptake plots of peptide aa411–419 from VDR H12 when heterodimer is bound to 1,25D3, DNA and SRC1 RID. **j** Differential deuterium uptake plots of peptide aa411–419 from VDR H12 when heterodimer is bound to Cmpd3, DNA, and SRC1 RID. The data are plotted as percent deuterium uptake verse time on a logarithmic scale

absence of these ligands, VBSs including DR3, DR3 half-site, *BGLAP* and *TRPV6*, and the RID of the co-activator protein SRC1. These studies provide snapshots of distinct conformational ensembles of VDRRXRα heterodimer in solution and allow multiple comparisons of sequential binding events (e.g., ligand, DNA, and co-activator protein) to probe alterations in the dynamics of the heterodimer to drive molecular interactions with co-regulators. While these HDX data do not provide specific orientations of each structural element within the heterodimer complex, they yield unique insight into local and global conformational fluctuations within the protein complex that could not be inferred from static structures.

Differential HDX analysis reveals unique intra-molecular and inter- molecular communications between heterodimer LBDs and DBDs in response to various VDR ligands and DNA VBSs. For instance, ligand binding to the LBD modulates dynamics of the remote zinc fingers within the DBD and DNA binding in turn enables allosteric perturbations impacting LBD dynamics. Although it is not known how the receptor hinge domain

interacts with the DBD and LBD, the HDX results support a structural model where the hinge plays a critical role in determining the structural arrangement of co-receptor's LBDs to DBDs. Various combinations of ligands and DNA VBSs stabilize the hinge of VDR to different extents, suggesting that it adopts unique conformations depending on the ligand and the specific sequence of the VBS. In the presence of 1,25D3, binding of DR3 half-site but not DR3 VBS alters receptor dynamics in the remote LBD. This contrasts with what is observed with synthetic agonist action wherein the binding of either DR3 or DR3 half-site VBSs drive analogous perturbations in LBD of VDR. Interestingly, a region spanning the hinge and H1 of the VDR LBD undergoes partial cooperative unfolding (EX1 kinetics) correlating with allosteric stabilizations observed in LBD suggesting the VDR hinge and H1 regions possess inherent flexibility enabling signals to be transmitted from the DBD to the LBD.

Interestingly, the presence of synthetic agonist and DNA drive long-range stabilization of hinge-H1 and H8–H9 loop region (part of the RXRα AF2 surface) in RXRα. Despite exhibiting

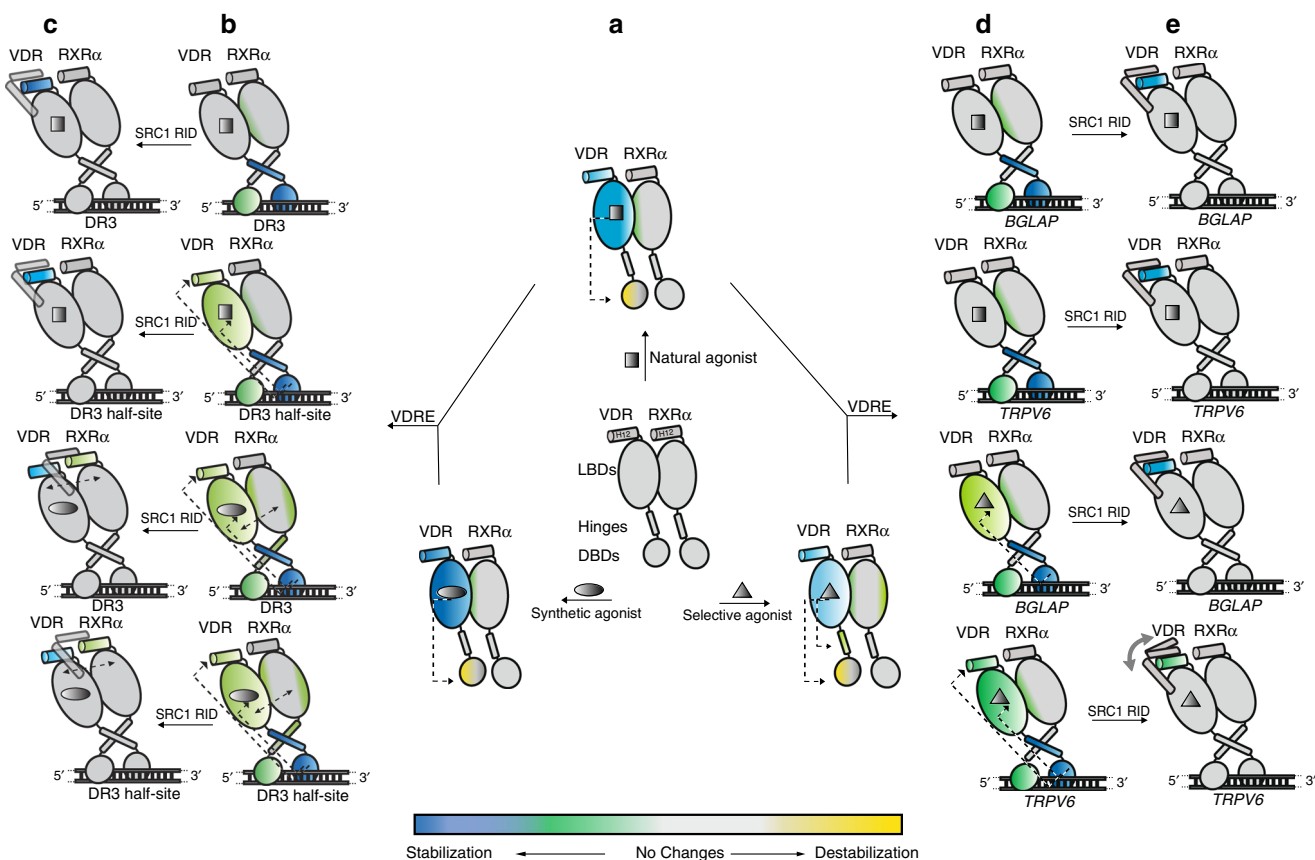

**Fig. 6** Schematic representation of VDRRXRα heterodimer activation by different ligand and DNA. Protein stabilization induced by interactions of the ligand, DNA and SRC1 RID (as measured by the differential HDX profiles) is shown in blue and green color, whereas destabilization is shown in yellow color. Regions with no significant change are colored gray. VDRRXRα heterodimer is shown with co-receptors' LBDs including AF2, hinge domains, and DBDs. Allosteric intra-molecular and inter-molecular communications within co-receptors are indicated by black dotted arrow. **a** Synthetic agonists; Cmpd's 1 and 2, induce greater stabilization in VDR LBD compared to that of a natural ligand 1,25D3. VDRM Cmpd3 exhibits relatively weak stabilization within AF2. Binding of all ligands induce allosteric destabilization within the VDR DBD. Additional stabilization was observed within the VDR hinge and H3 of RXRα upon binding to Cmpd3. The heterodimer interface becomes more stable upon ligand binding. **b** Binding of 1,25D3-bound heterodimer to DR3 and DR3 half-site result in stabilization within the DBDs and VDR hinge. Interaction with DR3 half-site leads to intra-domain allosteric stabilization in VDR LBD and AF2 that are not observed upon DR3 binding. In contrast, the presence of synthetic agonist and DNA with heterodimer drive stabilization in the DBD and hinge of both co-receptors and VDR AF2. **c** Large stabilization was observed in VDR AF2 upon binding SRC1 RID, whereas stabilization in RXRα AF2 was only observed when heterodimer is bound to synthetic agonist and DNA. **d** Cmpd3 could modulate heterodimer dynamics in an alternative mechanism, in which allosteric stabilization in VDR AF2 occurs upon heterodimer binding to *TRPV6* rather than *BGLAP*. In contrast, neither *BGLAP* nor *TRPV6* alter the conformational of VDR LBD when VDRRXR heterodimer is bound with 1,25D3. **e** Binding of SRC1 RID results in less stabilization at VDR AF2 when heterodimer is associated with Cmpd3 and *TRPV6*

relatively modest stabilization, the altered flexibility in these regions of RXRα may result in a unique orientation of the RXRα LBD relative to its dimer partner. This in turn would sequentially impact the positioning of co-activator upon interaction with the heterodimer perhaps impacting co-activator to DNA distance. In the context of the heterodimer, SRC1 RID binding drives stabilization of the AF2 surface in both VDR and RXRα. In contrast, binding of ligand and DNA to RXRα homodimer abrogates the perturbation dynamics of the RXRα LBD. This observation would suggest a potential role for VDR in mediating inter-receptor crosstalk within the heterodimer. Inter-molecular interactions would provide a direct mechanism for transducing information between the co-receptors VDR to RXRα.

As shown in Fig. 6, the results presented here suggest two distinct activation modes mediated by non-secosteroid synthetic agonists and a VDRM (Cmpd3) as compared to that observed with the natural ligand 1,25D3. The binding of VDRM to heterodimer leads to an unique stabilization of the VDR hinge that is not detected upon binding natural ligand or synthetic agonists, suggesting that ligand itself could have a direct effect on DNA binding. VDRM binding afforded less protection to solvent exchange within the LBP of VDR as compared to that observed upon binding 1,25D3. These differences in LBP dynamics that are initiated by ligand-binding properties likely give rise to the differences in DNA recognition properties and potency for SRC1 interaction observed for Cmpd3 when compared to 1,25D3. Binding of either *BGLAP* or *TRPV6* VBS to 1,25D3-bound heterodimer fails to perturb the dynamics of the LBDs and results in an open AF2 conformation susceptible for co-activator binding. Whereas binding of *TRPV6* but not *BGLAP* VBS to Cmpd3 liganded heterodimer specifically alters the stability of AF2 in VDR blunting the receptor's ability to interact with SRC1. It is important to note that the BGLAP and TRPV6 VBSs used in these studies are putative VDR response elements (VDREs) yet to be validated as bona fide VDREs in vivo. Taken together, these results suggest that the conformational flexibility of VDRRXRα heterodimer allows the core complex to adopt different conformations depending on the ligand bound, specific sequence of the VBS and co-regulatory protein binding, providing a common mechanism for ligand-dependency of NR activation on various target genes. The model presented may provide a framework for the design of improved VDMRs for the treatment of osteoporosis.

## Methods

**Reagents.** Full-length WT His-hVDR (Δ (151–190)) and WT Flag-h RXRα were expressed in a baculovirus system and purified by Ni-NTASEC or Flag-SEC, respectively. His-hSRC1 RID (627–786) variant 1 (NM_003743) was expressed in *Escherichia coli* (BL21 DE3) and purified using His-Trap (GE Healthcare). The final protein buffer was 50 mM Tris (pH 8.0), 150 mM NaCl, 10% glycerol and 2 mM DTT. The purity of each protein was > 95% and was verified using SDS-PAGE, western blot and MALDI MS.

The DR3 VBS 5′-CGTAGGTCAATCAGGTCACGTCGT-3′, DR3 half-site VBS 5′-CTAGCTCCCGAGGTCAGCGACGGCGCAGG-3′, *BGLAP* VBS 5′-CTAGGTGAATGAGGACAT-3′, *BGLAP* * VBS 5′-GGTGACTCACCGGGTGAACGGGGGCA-3′ and *TRPV6* VBS 5′-CAAGGGGTAGTGAGGTCAAAAGCA-3′ duplex oligonucleotides were purchased from Integrated DNA Technologies. Deuterium oxide (99.9 atom % D) was purchased from Sigma-Aldrich. Heterodimer complex was formed by mixing VDR and RXRα at 1:1 molar ratio (final concentration around 10 μM), and complex formation was confirmed by gel-shift assays. Vitamin D3 (Sigma) and synthetic Cmpds 1, 2, and 3 (provided by Eli Lilly, Indianapolis, IN, USA) were added at a 10-fold molar excess to heterodimer. SRC1 RID was added in a 2× molar ratio to heterodimer, and oligonucleotide (VBS) was mixed with the protein complex as needed (1.5× molar ratio). NMR data for the synthetic compounds are tabulated in the Supplemental Materials.

**Crystallography.** Human VDR LBD domain (119–425, Δ (166–216), S222L) with N-terminal his tag and SMT fusion was expressed in *E. coli*. The SMT fusion (7 mg) was cleaved with ULP1 enzyme (100 μl, 1 mg/ml) for 8 h at 4 °C with dialysis into buffer (20 mM Tris-HCl, pH 8.0, 500 mM NaCl, 10% glycerol, 25 mM imidazole, 0.2% BOG (b-octyl glucoside), 5 mM Bme (beta mercaptoethanol)).The

protein was purified by Ni-NTASEC and stored in final protein buffer 50 mM Tris (pH 8.0), 150 mM NaCl, 10% glycerol and 2 mM DTT. Crystals of the protein/Cmpd 3 complex (PDB code: 5V39) were grown by vapor diffusion with a well solution of 100 mM Tris-HCl pH 8.5, 25% PEG 3350 and 200 mM Ammonium Sulfate (reservoir buffer). Crystallization tray was set-up with 0.3 μl protein and 0.3 μl reservoir buffer (VDR LBD concentration was 3.1 mg/ml; ligand concentration 1 mM). The crystals had a space group of P212121 with cell dimensions of 44.00 Å, 52.67 Å and 105.88 Å. X-ray diffraction data were collected at beam line LRL-CAT at Advanced Photon Source (APS). The structures were solved by molecular replacement using a prior internal structure as a template. The structure of VDR LBD with Cmpd3 was determined to a resolution of 2.2 Å. The crystallographic refinement was done by Refmac 5[41] and Buster[42] while the model building was carried out by Coot[43]. The final refinement R-factors were $R_{work} = 0.167$, $R_{free} = 0.235$.

**Molecular docking.** All docking simulations were carried out in ICM Pro (Molsoft) using the Docking module. The crystal structure of the VDR LBD (5V39) with waters and ligands removed was used as the receptor. The ICM Pro software implements an algorithm of a biased probability Monte Carlo (BPMC) procedure (PMID: 8289329). Cmpds 1–2 were docked and the lowest energy poses were used for interpretation. Figures were generated in PyMol (Schrodinger).

**HDX-MS.** Solution-phase amide HDX experiments were carried out with a fully automated system (CTC HTS PAL, LEAP Technologies, Carrboro, NC; housed inside a 4 °C cabinet) as follows.

Peptide Identification: Peptides were identified using tandem MS (MS/MS) experiments performed with either a LTQ Orbitrap XL with ETD or a Q Exactive (Thermo Fisher Scientific, San Jose, CA) over a 70-min gradient. Product ion spectra were acquired in a data-dependent mode and the five most abundant ions were selected for the product ion analysis per scan event. The MS/MS *.raw data files were converted to *.mgf files and then submitted to MASCOT (version 2.3 Matrix Science, London, UK) for peptide identification. The maximum number of missed cleavages was set at 4 with the mass tolerance for precursor ions ± 0.6 Da and for fragment ions ± 8 p.p.m. Oxidation to methionine was selected for variable modification. Pepsin was used for digestion and no specific enzyme was selected in the MASCOT during the search. Peptides included in the peptide set used for HDX detection had a MASCOT score of 20 or greater. The MS/MS MASCOT search was also performed against a decoy (reverse) sequence and false positives were ruled out if they did not pass a 1% false discovery rate. The MS/MS spectra of all the peptide ions from the MASCOT search were further manually inspected and only the unique charged ions with the highest MASCOT score were included in HDX peptide set.

HDX-MS analysis: 10 μM of the apo protein was mixed with 1:10 molar excess of ligand and incubated for 2 h at 4 °C for complex formation before subjecting them to HDX analysis. For the differential HDX experiments, 5 μl of either the apo or the liganded protein complex with DNA and SRC1 RID were mixed with 20 μl of D₂O-containing HDX buffer (50 mM Tris, pH 8.0, 150 mM NaCl, and 2 mM DTT) and incubated at 4 °C for 0 s, 10 s, 30 s, 60 s, 300 s, 900 s, or 3,600 s. Following on-exchange, unwanted forward-exchange or back-exchange was minimized and the protein was denatured by the addition of 25 μl of a quench solution (1% v/v TFA in 3 M urea and 50 mM TCEP). Samples were then immediately passed through an immobilized pepsin column (prepared in house) at 50 μl min-1 (0.1% v/v TFA, 15 °C) and the resulting peptides were trapped and desalted on a 1.0 mm × 10 mm C₈ trap column (Hypersil Gold, Thermo Fisher, Grand Island, NY). The bound peptides were then gradient-eluted (5–50% CH3CN v/v and 0.3% v/v formic acid) across a 1.0 mm × 50 mm C₁₈ separation column (Hypersil Gold, Thermo Fisher, Grand Island, NY) for 6 min. Sample handling and peptide separation were conducted at 4 °C. The eluted peptides were then subjected to electrospray ionization directly coupled to a high resolution Orbitrap mass spectrometer (LTQ Orbitrap XL with ETD, Q Exactive, or Exactive, Thermo Fisher Scientific, San Jose, CA). Each HDX experiment was carried out in triplicate with a single preparation of each protein-ligand complex. The intensity weighted mean m/z centroid value of each peptide envelope was calculated and subsequently converted into a percentage of deuterium incorporation. This is accomplished by determining the observed averages of the undeuterated and fully deuterated spectra using the conventional formula described elsewhere[44]. In the absence of a fully deuterated control, 100% deuterium incorporation was calculated theoretically, and corrections for back-exchange were made on the basis of an estimated 70% deuterium recovery and accounting for 79.9% final deuterium concentration in the sample (1:5 dilution in D₂O HDX buffer). Statistical significance for the differential HDX data is determined by an unpaired t-test for each time point, a procedure that is integrated into the HDX Workbench software[45].

Data Rendering: The HDX data from all overlapping peptides were consolidated to individual amino acid values using a residue averaging approach. Briefly, for each residue, the deuterium incorporation values and peptide lengths from all overlapping peptides were assembled. A weighting function was applied in which shorter peptides were weighted more heavily and longer peptides were weighted less. Each of the weighted deuterium incorporation values were then averaged incorporating this weighting function to produce a single value for each amino

acid. The initial two residues of each peptide, as well as prolines, were omitted from the calculations. This approach is similar to that previously described[46].

Data Statistics: Deuterium uptake for each peptide is calculated as the average of % D for all on-exchange time points and the difference in average %D values between the apo and ligand bound samples is presented as a heat map with a color code given at the bottom of the figure (warm colors for deprotection and cool colors for protection). Peptides are colored by the software automatically to display significant differences, determined either by a > 5% difference (less or more protection) in average deuterium uptake between the two states, or by using the results of unpaired $t$-tests at each time point ($p$-value < 0.05 for any two time points or a $p$-value < 0.01 for any single time point). Peptides with non-significant changes between the two states are colored gray. The exchange at the first two residues for any given peptide is not colored. Each peptide bar in the heat map view displays the average Δ %D values, associated standard deviation, and the charge state. Additionally, overlapping peptides with a similar protection trend covering the same region are used to rule out data ambiguity.

**Homology modeling**. A structural model of the VDR (aa 18–455)/RXRα (aa 135–462) heterodimer bound to the DR3 VBS DNA fragment was created using Molsfot ICM Pro software. Amino Acids 201–223 of RXRα and 166–216 of VDR could not be modeled due to the lack of available homology models containing these amino acid residues. The heterodimer was created in the "open" conformation and a partial cryo-EM model (generous gift from Professor Bruno Klaholz and Dino Moras) was used to guide the creation of the model. Loops that could not be modeled from the cryo-EM data were added (mostly in the hinge regions) and hydrogen atoms were added to all amino acids and both ligands. The added loops were subject to loop modeling to improve clashing using the ICM-Pro loop modeling utility. The final structure was subjected to minimization, regularization, and annealing in ICM Pro.

**BGLAP and TRPV6 activation assays**. Human osteosarcoma cells MG-63, (CRL-1427, American Type Culture Collection (ATCC), Manassas, VA), were maintained in EMEM (ATCC 30–2003) + 10% FBS in 5% CO₂. For bone gamma-carboxyglutamate protein (*BGLAP*) expression analysis MG-63 cells, suspended in EMEM (ATCC 30–2003) + 5% charcoal dextran treated FBS, were seeded onto 96-well tissue culture treated plates at 25,000 cells per well. After an overnight incubation the medium was removed and serial dilutions of VDRMs in concentrations ranging from 10 μM to 2 nM in EMEM (ATCC 30–2003) + 5% charcoal dextran treated FBS were added. Following a 24 h incubation the medium was removed and the plates were sealed and stored at −80 °C for later processing. QuantiGene 2.0 Assay Kit (Affymetrix QS0009) was used to quantitate *BGLAP* and the House-keeping gene GUSB mRNA levels. Following the kit instructions, lysis buffer was added to the frozen cells and incubated briefly at 55 °C. A volume of 20 μl of BGLAP probeset (Affymetrix SA-10875-01) or 18 S (Affymetrix SA-10026) in blocking buffer was added to the provided capture plates before adding 80 μl of the cell lysate. The plates were sealed and incubated at 55 °C overnight. Following an overnight incubation the capture plates were washed and the probes amplified following kit instructions. The resulting luminescence signal was detected on an Envision (Perkin Elmer). *BGLAP* signal was normalized using the 18S signal and the resulting data was fit to a 4-parameter logistics to determine EC₅₀.

C2BBe1 (a clone of CaCO₂) cells (CRL-2101, American Type Culture Collection, Manassas, VA) were maintained in DMEM (ATCC 30-2002) supplemented with 10% FBS and insulin-transferrin-selenium-G supplement (Invitrogen, Carlsbad, CA). For TRPV6 expression analysis, cells were plated in 96-well tissue culture treated plates (40,000 per well) in differentiation medium (DMEM containing 5% charcoal-stripped FBS and insulin-transferrin-sodium selenite supplement (ITS)). Cells were allowed to differentiate for 6 days with medium replacement every other day before compound treatment. The medium was removed, and serial dilutions of 1,25D3 or VDRMs in concentrations ranging from 10 μM to 2 nM in differentiation medium were added. After 24 h, the medium was removed, and cell plates were sealed and frozen at −80 °C until assayed. QuantiGene 2.0 Assay Kit (Affymetrix QS0009) was used to quantitate TRPV6 and housekeeping gene GUSB mRNA levels. Following the kit instructions, lysis buffer was added to the frozen cells and incubated briefly at 55 °C overnight. Following an overnight incubation the capture plates were washed and the probes amplified following kit instructions. The resulting luminescence signal was detected on an Envision plate reader (Perkin Elmer). TRPV6 signal was normalized using the GUSB signal and the resulting data was fit to a 4-parameter logistic to determine EC50 values.

**Gel retardation analysis**. Full-length VDRRXRα heterodimer (10 μM, 20 μl) in protein buffer was incubated with respective VBS (2X molar ratio) or anti VDR antibody (5 μl, 200 μg/ml, sc-13133, Santa Cruz Biotechnology) for 30 min on ice before mixing with 5 μl Hi-Density TBE sample buffer (Invitrogen). These reactions were analyzed on a 6% DNA Retardation gel (Invitrogen) in 0.5X TBE buffer. Electrophoresis was performed at a constant voltage of 150 V at room temperature. Gel was stained with ethidium bromide and coomassie blue for DNA and protein visualization, respectively.

**Data availability**. Further data supporting the findings of this study are available from the corresponding author upon reasonable request. The accession code for VDRM bound VDR LBD is 5V39 (Protein Data Bank).

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

## Acknowledgements

The crystallography work used resources of the Advanced Photon Source, a U.S. Department of Energy (DOE) Office of Science User Facility operated for the DOE Office of Science by Argonne National Laboratory under Contract No. DE-AC02-06CH11357. We thank Kris Conners, Milan Maletic for the cloning and fermentation work, Brad Condon for the purification and Laura Pelletier for crystallization in the VDR crystallography. We thank Robert J. Barr for screening the VDRMs in BGLAP and TRPV6 cellular assays.

## Author contributions

J.Z., R.E.S., M.J.C., J.A.D., and P.R.G. conceived of the project and designed the research; J.Z., M.R.C., R.E.S., Y.W., S.J.N., R.D.G.-O. conducted the research. J.Z., M.R.C., R.E.S., Y.W., J.B.B., B.D.P., K.R.S., M.J.C., J.A.D., and P.R.G. analyzed the data. J.Z., R.E.S., J.A.D., and P.R.G. wrote the paper with contributions from all authors.

## Additional information

**Competing interests:** The authors declare no competing financial interests.

