## [Peer Review file · Nature Communications]

Reviewers' comments:

Reviewer #1 (Remarks to the Author):

This manuscript describes that conformational dynamics controlling promoter specific VDR target gene activation was revealed by HDX mass spectra. Extremely high quality and quantity experiments were performed and provided results and conclusions are clear cut. Specifically, authors showed that 1,25D3, synthetic nonsteroidal agonist, and VDRM binding stabilizes VDR and RXR but the stabilization mechanisms are different from each other in terms of structural-biological viewpoint, resulting in the distinct actions of these compounds. Complicated mechanisms were clearly shown by author's high techniques and insights. This reviewer does not have any major issues with the work.

Minor comments

Page 8, last line: Spell of "kick residue" is correct?

Page 9, line 11: VDRE is an abbreviation of vitamin D response element. DNA response element is incorrect.

Page 11, line 8: TRPV6 is correct? Authors should correct TRPV6 to CYP24A1.

Page 19: line 3: -80C should be corrected to -80°C.

Page 24, line 14: "absence of compd 3" is correct? Please check this sentence.

Throughout the manuscript, both AF2 and AF-2 are used. Please unify.

Reviewer #2 (Remarks to the Author):

The manuscript by Zheng et al. presents the results of an extensive biophysical/HDX-MS study characterizing the interaction between the VDRXR α heterodimer and both small molecule endogenous and synthetic ligands, several target DNA sequences and a co-regulatory protein. VDRXR α is a significant drug target for treatment of osteoporosis and thus an analysis of its interaction with drugs and endogenous binding partners is of broad interest. The authors characterize the interactions of the VDR ligand binding domain (LBD) to small molecular ligands by both X-ray crystallography and HDX, while the interactions of the intact VDRXR α heterocomplex with VDR target DNA sequences and the co-regulatory SRC1 protein is investigated solely by HDX-MS.

Overall, the manuscript is well written and provides high quality experimental data on a complex, delicately regulated, protein system. To the mind of this reviewer, the earlier reports of crystal structures of the VDR LBD bound to the natural ligand (1,25D3) and several other non-secosteroids however detracts somewhat from the general novelty of the reported crystal structure. Also, the authors have previously used HDX-MS to study the intact VDRXR α (Nat Struct Mol Biol. 2011 May;18(5):556-63. doi: 10.1038/nsmb.2046. Epub 2011 Apr 10), mapped the conformational effects of agonist binding and reported that DNA binding to VDRXR α induces long-range allostery effects. However, the new HDX-MS data in this report is significantly more comprehensive by including several new functional states not probed before, is acquired with evident expertise and represents a considerable effort. Importantly, the data provides new mechanistic insights into how DNA, endogeneous co-activators (SRC1-RID) and synthetic agonists may regulate complex conformational events that impact the function of the nuclear receptor complex. Apart from HDX-MS, there are few other sensitive methods available that can directly compare the conformational properties in solution of this wide range of functionally-relevant states of the receptor. As such, I am positive towards publication. I have some specific concerns that should be addressed in order to improve the manuscript.

Major comments:

- The readability of the manuscript introduction should be improved. Suggestions include:
 - A) Short sentences early in the introduction describing the target biology, for instance, what is bone mineralization and the fact that 1,23D3 is the active metabolite of vitamin D3 will ease the understanding of the manuscript - especially for a reader unfamiliar with this biology.
 - B) A schematic figure of the complexes would be highly beneficial to guide the reader and facilitate the understanding of the presented data and investigated complexes. Specifically all the areas in VDR as described the second paragraph of the introduction.
 - C) Many abbreviations are used before they are introduced and explained, in particular making the abstract a hard read.
 - o VDRXR α
 - o AF2
 - o SRC1
 - o NR
- - I have several concerns regarding the presented EX1 kinetic data on Supplementary Figure 2 d.
 - o The low mass population in an EX1 time-regime will decrease over time. This is however not seen for VDRXR α :1.25D3 + CYP24A1, where the low mass population at T=30s is markedly larger than at T=10s. Similar observations can be seen for VDRXR α :cmpd2 + DR3, where the low mass population at T=900 is larger than that at all prior time points. The authors need to examine this carefully and address this discrepancy. Have other reasons for the appearance of two different envelopes been considered?
 - o When showing peptides undergoing exchange according to EX1 kinetics, it can be very helpful to show the deconvolution of the two different envelopes (low mass and high mass). These curves can for example be generated by softwares like HX-Express. Also, how did the authors calculate deuterium incorporation for peptides showing EX1 kinetics.
 - o Spectra on supplementary figure 2d should be properly aligned.
 - When making quantitative comparisons of changes in HDX induced by different ligands on the receptor, a similar receptor occupancy of the compared complexes should be ensured. The authors should describe how this was ensured.

Minor comments:

- P.4 "SRC-1), These transcriptional coregulatory proteins contain conserved helical nuclear receptor box (NR box) motifs 5' - LXXLL - 3'. This sentence is not clear.
- A small color bar on every figure defining the effects would be helpful to the reader.
- The authors write "The Corrections for back exchange were made on the basis of an estimated 70% deuterium recovery and accounting for the known 80% deuterium content of the ion-exchange buffer." The authors should outline have they have estimated the back-exchange and report the average value and the range.
- Minor typographic mistakes were observed:
 - o "Hydrogen/deuterium exchange coupled with mass spectrometry HDX-MS has been shown to be a robust biophysical method to probe solution phase protein conformational dynamics in the context of ligand and protein-protein interactions 9,21-25 Here we". A period is missing.
 - o " In addition, the VDR hinge/H1 peptide (aa109-133) underwent partial unfolding (as revealed by observing EX1 kinetics) when heterodimer was bound to synthetic agonists and DNA (Supplementary Fig. 2e)". Wrong reference – should be fig. 2D.
 - o Supplementary Figure 2b -The sequence is very difficult to read.

Reviewer #3 (Remarks to the Author):

Reviewers: Matthew Knuesel and Keith Yamamoto

In an elegant application of hydrogen-deuterium exchange mass spectrometry, Zheng et al. demonstrate here how multiple signaling inputs (including DNA, ligand, and coregulator peptide) act combinatorially to selectively modulate structural dynamics and solvent accessibility of the VDRXRa heterodimer. They discriminate short- and long-range conformational perturbations conferred upon VDRXRa by the input signals individually and in combination. Framed by this foundational platform, the authors identify a selective VDR modulator (VDRM), Cmpd3; determine the VDR LBD-Cmpd3 structure by X-ray crystallography; model the VDR-LBD interaction with two other synthetic ligands; analyze by HDX MS the VDRXRa:Cmpd3 complex bound to two different VDR DNA binding sequences linked to 1,25D3-regulated genes important for bone formation and hypercalcemia; and show that these genes are regulated differentially by Cmpd3.

Against a matrix of VDR binding sites, ligands and a coregulator peptide, Zheng et al. compare VDRXRa HDX-MS profiles under 28 distinct signaling conditions, demonstrating the conformational malleability of the receptor, with particular note to the extent and range of allosteric crosstalk. For example, binding to different DNA sequences alters HDX kinetics extending from the DBDs through the hinges and most extensively in the AF2 regions of the LBDs; similarly, specific ligand interactions trigger alterations in structural dynamics that extend into the DBDs, and changes in one heterodimer partner produce structural consequences in the other.

This comprehensive study will be broadly impactful for those invested in VDR mechanism and drug design, as well as transcriptional regulation, protein dynamics, structure, and allostery. In general, the work is clearly and logically presented, although there are several points that the authors should address prior to acceptance for publication:

Major points:

1. The authors extend a persistent failure in the field to define and distinguish properly the DNA and genomic components relevant to transcription and transcriptional regulation. These failures produce confusion, and more importantly misinterpretations and overinterpretations, producing misconceptions that inhibit progress. Corrections should be made throughout the manuscript, adopting the nomenclature and definitions below:

Vitamin D Response Element (VDRE): a genomic segment that confers a particular VDR response in vivo. In the present work, in vivo regulatory activity has not been validated for any of the investigated DNA segments, so VDRE is in no case appropriate terminology.

VDR Binding Sequence (VBS): a DNA sequence bound specifically, with high affinity, by VDR in vitro. A VBS has no ascribed function in vivo, either for VDR binding or for transcriptional regulation. The present work examines exclusively VBSs, none of which has been tested or validated as a VDRE; this should be corrected throughout.

VDR Occupied Region (VOR): a genomic segment occupied selectively by VDR in vivo. A VOR, typically detected as a "ChIP seq peak", carries no implication of transcriptional regulatory activity. In the present work, VORs are denoted as VDREs, as in "VDRXRa translocates and associates with high binding affinity to genomic VDR response elements (VDREs)" (page 4). This should be corrected throughout; it would be appropriate to note that some of the VBSs studied are taken from VORs (although none of the VORs has been validated as a VDRE).

Promoter: a genomic segment at which general transcription factors, such as TFIIB, and RNA polymerase bind and initiate transcription of functional RNAs. Response elements, which bind transcriptional regulatory factors, such as VDR, rarely overlap even partially with promoters

(typically they are separated by 10-100kb or more), and the two entities are functionally distinct. In the present work, definitions of VDRE and promoter are conflated; this should be corrected throughout, including in the manuscript title.

General issues surrounding these definitions and functions are considered by Weikum et al. (Nature Rev Mol Cell Biol (2017) 18:159–74. Two authors of that perspective are responsible for this review, and would be happy to speak directly with Dr. Griffin and his coworkers if that would be useful in finalizing their manuscript.

2. Page 11: "The VDREs within the promoter region of BGLAP (5'-CTAGGTGAATGAGGACAT-3') and TRPV6 (5'- CAAGGGGTAGTGAGGTCAAAGCA-3') were chosen as they contain two direct repeats separated by a three base-pair gap, a signature highly reminiscent of canonical VDR response element 33,34." This statement exemplifies some of the concerns noted above: the "signature" describes a family of closely related VBSs, i.e., a motif, not a "canonical VDR response element"; moreover, these are not part of "the promoter region". The authors should state whether the VBSs tested are from genomic regions that are VDR occupied in vivo (i.e., are within VORs) in cells where the candidate target genes are VDR-responsive; the positions of selected VORs relative to candidate target genes should be provided. Notably, VBSs within the genome greatly outnumber VORs, which in turn outnumber VDREs; VORs differ in different cells, and VDREs can operate at very long range. Hence, proximity of VBSs or VORs to VDR-regulated genes identifies neither VDREs nor their target genes; this should be clarified/corrected in Table 1 and throughout the manuscript. Validation of a VOR and resident VBS as a VDRE, and of a candidate target gene as a bona fide target gene, would require genome editing that produces allele-specific loss of regulation of the candidate target gene. It is of course interesting that VDRRXR α binding to the selected VBSs provokes alterations in HDX kinetics. The reviewers accept that the authors may consider such validating CRISPR/Cas9 mutagenesis studies to be outside the scope of their already expansive investigation. If this is the case, they should state clearly that functional validation beyond VBS has not been done, and they should correct nomenclature accordingly.

3. A BLAT search of each of the VBSs used in this study in the human genome (UCSC GRCh38/hg38) and BLAST through NCBI brought these results:

DR3 (5'-CGTAGGTCAATCAGGTACGTCGT-3'): This sequence appears to be a synthetic construct composed of canonical binding motifs separated by a ATC spacer and flanked by 1-2 CTG repeats. It does not match any sequence in the human genome browser (UCSC genome browser human genome GRCh38/hg38). It should be explained as such. DR3 should not be italicized as it is throughout the text, as DR3 is not a gene name, but a synthetic DNA sequence "direct repeat with a 3 bp spacer".

CYP24A1 (5'-CTAGTCCCGAGGTCAGCGACGGCGCAGG-3'): No exact matches to this sequence were found within the human genome (UCSC) or in a BLAST is used on the Human Genomic + Transcript database (NCBI). Perhaps there is an altered region flanking a core VBS? Please explain.

BGLAP (5'-CTAGGTGAATGAGGACAT-3'): This sequence is not found in UCSC genome browser human (GRCh38/hg38) or when BLAST is used on the Human Genomic + Transcript database. Please explain.

TRPV6 (5'-CAAGGGGTAGTGAGGTCAAAGCA-3'): This sequence is located ~4.3 KB upstream from the transcription start site of TRPV6 and underlies a VOR identified by ChIP. Meyer et al. (Mol Endo (2006) 20:1447–61) showed that this fragment, as well as numerous others underlying VORs proximal to TRPV6, confer VDR-mediated regulation in a reporter assay. Such plasmid-based reporter assays are not equivalent to genomic alterations introduced by Cas9-based editing, and are not sufficient to identify a functional VDRE. The authors may wish to denote them as "candidate VDREs", although "VBSs" may be more accurate for this study.

4. The transcriptional data presented in Table 1 would be better displayed as a figure, with a better explanation of how EC50 and %MaxStim were determined (along with measures of error in each experiment). This could be potentially be incorporated into Fig. 1.

5. The HDX-MS “stabilization – destabilization” scale is shown only in the Supplement. As this is central to all analyses, it should be included and explained briefly in Fig. 2 (and referenced in Figs 3-5).

Minor points:

6. Figures contain ligands and DNA that are colored in hues that overlap with the HDX-MS scale, rendering it difficult to differentiate what is VDRXRα undergoing conformational changes and what is DNA or ligand (particularly for the orange ligand displayed in Figs. 2 and 5). Recoloring the ligands and DNA to be distinct from the HDX-MS range would facilitate interpretation of the results.

7. In the Supplement, the authors consider different VDRXRα models derived from X-ray crystallography and from cryo-EM/SAXS. It would be helpful to mention more prominently these models in the main text, and to discuss the “fit” of each to the HDX-MS data.

8. It would be helpful to expand the explanation of EX1 and EX2 kinetics beyond “occurrence of EX1 kinetics in protein native state has been shown to provide important clues to protein intermediate conformational states in solution^{22,39,40}”.

9. The rationale for displaying some model structures in Figs. 2 - 5 in ribbon and some in surface is presumably to differentiate SRC1-bound complexes from those lacking SRC1. In any case, some explanation should be added to the legends. In addition, the surface renderings would benefit from an outline of the boundaries of each protein, so readers could more readily discern which receptor surfaces display altered HDX-MS dynamics.

10. Simple edits:

Capital C in Cmpd throughout.

Fig. 1d legend asserts that “VDR LBD is shown in ribbons” whereas ribbon is not shown and side chains are displayed in ball and stick.

Figs. 2-5, each legend: “Percentages of deuterium differences are color-coded according to the smooth color gradient key. Dark gray, no statistically significant changes between compared conditions; light gray, regions that have no sequence coverage and include proline residue that has no amide hydrogen exchange activity.” Include this statement in Fig. 2 legend only; for Figs. 3-5, use “Percentages of deuterium differences are coded as in Fig 2.”

Ensure that Methods section is consistent in terminology and nomenclature. In the present version, for example, the Reagents subsection cites Ni-NTASEC and denotes a deletion as “Δ”, whereas the Crystallography subsection refers to “Nickel chelate affinity chromatography and size-exclusion chromatography”, and denotes a deletion as “del.”

Reviewers' comments:

Reviewer #1 (Remarks to the Author):

This manuscript describes that conformational dynamics controlling promoter specific VDR target gene activation was revealed by HDX mass spectra. Extremely high quality and quantity experiments were performed and provided results and conclusions are clear cut. Specifically, authors showed that 1,25D3, synthetic nonsteroidal agonist, and VDRM binding stabilizes VDR and RXR but the stabilization mechanisms are different from each other in terms of structural-biological viewpoint, resulting in the distinct actions of these compounds. Complicated mechanisms were clearly shown by author's high techniques and insights. This reviewer does not have any major issues with the work.

Authors' Response:

We thank the reviewer for their positive comments about the quality and importance of our manuscript; "Extremely high quality and quantity experiments were performed and provided results and conclusions are clear cut."; and "Complicated mechanisms were clearly shown by author's high techniques and insights."

Minor comments

Page 8, last line: Spell of "kick residue" is correct?

Authors' Response:

We apologize for this typographical error. We have edited "kick" to "kink residue" in the text.

Page 9, line 11: VDRE is an abbreviation of vitamin D response element. DNA response element is incorrect.

Authors' Response:

We have address this error and address nomenclature issues also raised by Reviewer 3 throughout the manuscript.

Page 11, line 8: TRPV6 is correct? Authors should correct TRPV6 to CYP24A1.

Authors' Response:

Thank you for pointing out this error. We have also modified the description of this oligonucleotide as a synthetic half-site DR3 (DR3 half-site) which was designed in part based on a VBS within the CYP24A1 promoter. We have corrected this nomenclature throughout the text.

Page 19: line 3: -80C should be corrected to -80°C.

Authors' Response:

Corrected.

Page 24, line 14: "absence of cmpd 3" is correct? Please check this sentence.

Authors' Response:

This sentence has been modified as: "additional stabilization was observed within the VDR hinge and H3 of RXR α upon binding to Cmpd3."

Throughout the manuscript, both AF2 and AF-2 are used. Please unify.

Authors' Response:

Thank you. We have unified it as AF2.

Reviewer #2 (Remarks to the Author):

The manuscript by Zheng et al. presents the results of an extensive biophysical/HDX-MS study characterizing the interaction between the VDRRXR α heterodimer and both small molecule endogenous and synthetic ligands, several target DNA sequences and a co-regulatory protein. VDRRXR α is a significant drug target for treatment of osteoporosis and thus analysis of its interaction with drugs and endogenous binding partners is of broad interest. The authors characterize the interactions of the VDR ligand binding domain (LBD) to small molecular ligands by both X-ray crystallography and HDX, while the interactions of the intact VDRRXR α heterocomplex with VDR target DNA sequences and the co-regulatory SRC1 protein is investigated solely by HDX-MS.

Overall, the manuscript is well written and provides high quality experimental data on a complex, delicately regulated, protein system. To the mind of this reviewer, the earlier reports of crystal structures of the VDR LBD bound to the natural ligand (1,25D3) and several other non-secosteroids however detracts somewhat from the general novelty of the reported crystal structure. Also, the authors have previously used HDX-MS to study the intact VDRRXR α (Nat Struct Mol Biol. 2011 May;18(5):556-63. doi: 10.1038/nsmb.2046. Epub 2011 Apr 10), mapped the conformational effects of agonist binding and reported that DNA binding to VDRRXR α induces long-range allostery effects. However, the new HDX-MS data in this report is significantly more comprehensive by including several new functional states not probed before, is acquired with evident expertise and represents a considerable effort. Importantly, the data provides new mechanistic insights into how DNA, endogeneous co-activators (SRC1-RID) and synthetic agonists may regulate complex conformational events that impact the function of the nuclear receptor complex. Apart from HDX-MS, there are few other sensitive methods available that can directly compare the conformational properties in solution of this wide range of functionally-relevant states of the receptor. As such, I am positive towards publication. I have some specific concerns that should be addressed in order to improve the manuscript.

Authors' Response:

We thank the reviewer for their positive comments on the manuscript.

Major comments:

- The readability of the manuscript introduction should be improved. Suggestions include:
A) Short sentences early in the introduction describing the target biology, for instance, what is bone mineralization and the fact that 1,23D3 is the active metabolite of vitamin D3 will ease the understanding of the manuscript - especially for a reader unfamiliar with this biology.

Authors' Response:

We fully appreciate the comments on “readability” and we have attempted to address this as follows:

We have made edits in the introduction to describe what the bone mineralization process is, and we have added additional information about 1,25D3: “an active metabolite of vitamin D3.”

Here is the updated text region:

Calcium ion metabolism and homeostasis is cooperatively governed by the intestine, kidney, and bone to ensure physiological bone mineralization, which is an important process of laying down calcium phosphate on bone matrix¹. The vitamin D receptor (VDR), a member of the nuclear receptor superfamily, orchestrates calcium homeostasis and bone mineralization through transcriptional control of VDR target genes in various tissues. VDR is activated by the full agonist secosteroid hormone 1,25D3^{2,3} (also known as an active metabolite of vitamin D3) resulting in increased expression of the osteoblast hormone osteocalcin, or bone gamma-carboxyglutamic acid-containing protein (*BGLAP*) which is a protein essential for bone formation⁴.

B) A schematic figure of the complexes would be highly beneficial to guide the reader and facilitate the understanding of the presented data and investigated complexes. Specifically all the areas in VDR as described the second paragraph of the introduction.

Authors' Response:

We fully agree. We have listed all the functional complexes/states that were investigated in this study within **Supplementary Figure 1a** as a guide for readers. Further to this point, we further refined the full length VDRXRα heterodimer model using homology modeling based on cryo-EM VDRXRα bound to DR3 structure. We present this model in **Supplementary Figure 2c** and we illustrate the domain arrangement of VDR and RXRα within the heterodimer when complexed with DR3 VBS (VDR binding sequence). As shown below, we put relevant descriptions of the current model in the below paragraph to improve the readability of the manuscript.

Supplementary Figure 2c: VDR/RXR α homology model

Legend: VDR/RXR α DR3 model: the VDR/RXR α heterodimer was created in the “open” conformation and a partial cryo-EM model (generous gift from Professor Dino Moras) was used to guide the creation of the model. VDR/RXR α heterodimer forms an extended, L-shaped organization with RXR α DBD occupying one half-site of DR3 at 5'- end and VDR DBD occupying another at 3'- end^{5,6}. VDR hinge domain, as a connection linker between DBD and LBD, plays a key role in determining structural orientation of LBD dimer, which is arranged perpendicular to DR3. Residue Pro122, which resides on the VDR hinge domain, was described as the “kink residue” between the C-terminal hinge helix and helix H1 of the LBD playing an important role in dictating the orientation of the LBD, resulting in an open LBD dimer architecture facing away from DBD. In this model, VDR hinge domain adopts an α -helical structure residing closely to DR3 and makes extensive contacts with phosphate and backbone of DNA⁶. This is in consistency with our HDX observations that VDR hinge undergoes different extent of protections upon binding to various VBSs. Unlike VDR, DNA binding does not perturb RXR α hinge dynamics as it forms a flexible linker that allows enhanced adaptability of heterodimer to diverse response elements. This model suggests that the hinges could possess intrinsic properties to orient LBD dimer in a precise way and provide structural basis for crosstalk between LBDs and DBDs.

C) Many abbreviations are used before they are introduced and explained, in particular making the abstract a hard read.

Authors' Response:

We have modified and updated the nomenclature and appearance of each abbreviation used in the abstract.

o VDRXR α - vitamin D receptor/retinoid X receptor- α

o AF2 – activation function 2

o SRC1 - steroid receptor coactivator-1

o NR – nuclear receptor

o SRC1-RID - receptor interaction domain (RID) of the p160 NR co-activator steroid receptor co-activator 1

• I have several concerns regarding the presented EX1 kinetic data on Supplementary Figure 2d.

o The low mass population in an EX1 time-regime will decrease over time. This is however not seen for VDRXR α :1, 25D3 + CYP24A1, where the low mass population at T=30s is markedly larger than at T=10s. Similar observations can be seen for VDRXR α :cmpd2 + DR3, where the low mass population at T=900 is larger than that at all prior time points. The authors need to examine this carefully and address this discrepancy. Have other reasons for the appearance of two different envelopes been considered?

Authors' Response:

We thank the reviewer for their careful attention to the data presented on EX1 behavior. As suggested, we carefully reanalyzed the data using HX-express 2 software^{7,8}. We feel we have now address any discrepancies and provide a new **Supplementary Figure 2d** showing output from HX Express. The analysis of high and low mass envelopes by HX express suggests the presence of structurally heterogeneous states of VDR in the region of this peptide (KRKEEEALKDSLRLPKLSEEQRIIA, +4). We have also analyzed the peak width of all spectra showing EX1 and EX2 kinetics. The peak width analysis is a robust and quick way of characterization of EX1 kinetics⁷. Spectra undergoing EX1 kinetics have much wider peak width compared to EX2 kinetics (shown now in **Supplementary Figure 2d, lower panel**).

It is also possible that the H1 region of the VDR hinge domain intermittently interacts with DNA resulting in cooperative exchange events in several residues that have not formed stable interaction with DNA. It is also possible that VDR hinge-H1 region adopts two different secondary structural elements – loop-like structure at its N-terminus (hinge loop region) and helix structure at its C-terminus (beginning of Helix 1). These two different structural elements would confer distinctive protection factors that could give rise to two distinct populations if the difference between protection factors are large enough.

o When showing peptides undergoing exchange according to EX1 kinetics, it can be very helpful to show the deconvolution of the two different envelopes (low mass and high mass). These curves can for example be generated by software like HX-Express. Also, how did the authors calculate deuterium incorporation for peptides showing EX1 kinetics.

Authors' Response:

We fully agree and we have analyzed the EX1 spectra by HX express 2 and plotted relative deuterium level of high and low MS envelopes (**Supplementary Figure 2d, middle panel**).

o Spectra on supplementary figure 2d should be properly aligned.

Authors' Response:

We now have carefully aligned the spectra produced by HX express 2 software.

• When making quantitative comparisons of changes in HDX induced by different ligands on the receptor, a similar receptor occupancy of the compared complexes should be ensured. The authors should describe how this was ensured.

Authors' Response:

We are in full agreement with this comment. As described in the Methods Section, Cmpd's 1, 2, and 3 were added to the VDRXR α heterodimer at a tenfold molar excess (100uM) to ensure saturation ligand binding to VDR.

Minor comments:

• P.4 "SRC-1), These transcriptional coregulatory proteins contain conserved helical nuclear receptor box (NR box) motifs 5' - LXXLL - 3'. This sentence is not clear.

Authors' Response:

We agree and we have re-written the sentence as follows: "These transcriptional coregulatory proteins contain conserved helical nuclear receptor box (NR box) motifs 5' - LXXLL - 3', which is required for the ligand dependent binding of transcriptional activator to nuclear receptor⁹."

• A small color bar on every figure defining the effects would be helpful to the reader.

Authors' Response:

We apologize for this oversight. We have updated all relevant figures with a HDX color key.

• The authors write "The Corrections for back exchange were made on the basis of an estimated 70% deuterium recovery and accounting for the known 80% deuterium content of the ion-exchange buffer." The authors should outline have they have estimated the back-exchange and report the average value and the range.

Authors' Response:

We have corrected the text to read "79.9 % deuterium content in the on-exchange buffer."

Estimate of 70% deuterium recovery: We routinely run a maximum deuterated sample on each of our HDX systems to monitor the overall performance of the system. Here we report the results from such a maximum deuterated sample run recently on the system. It is about 70% Average deuterium recovery, 41%-90% range (Figure 2). The two lower recovery peptides contain multiple His. While our platform is not fully maximized to reduce deuterium loss post quench, the sample is maintained at low pH and all of the solvents, syringes, valves, columns (except the protease column which is at 15°C) are maintained at 4°C within a deli frig. The transfer tube from the deli frig to the mass spectrometer is as short as possible and insulated, and the ion source conditions are carefully set to balance deuterium loss with ion signal. With this set up we routinely obtain 70% Average, 50%-90% range deuterium recovery. Regardless, we have shown in a recent publication the reproducibility of this platform¹⁰. And given that all of the experiments are differentials run within the same day, the deuterium loss is expected to be equivalent for peptides from condition A versus condition B.

Percentage of deuterium recovery

While 70% is an estimation and the percentage varies from peptide to peptide, we use this as a correction factor with HDX Workbench software¹¹. Thus, we believe this is a good approach for differential HDX experiments were both samples are treated identically (same LC gradient, pH 2.4, 0°C).

• Minor typographic mistakes were observed:

o "Hydrogen/deuterium exchange coupled with mass spectrometry HDX-MS has been shown to be a robust biophysical method to probe solution phase protein conformational dynamics in the context of ligand and protein-protein interactions 9,21-25 Here we". A period is missing.

Authors' Response:

The sentence has been modified as: "Hydrogen/deuterium exchange coupled with mass spectrometry HDX-MS has been shown to be a robust biophysical method to probe solution phase protein conformational dynamics in the context of ligand and protein-protein interactions in past decade".

o " In addition, the VDR hinge/H1 peptide (aa109-133) underwent partial unfolding (as revealed by observing EX1 kinetics) when heterodimer was bound to synthetic agonists and DNA (Supplementary Fig. 2e)". Wrong reference – should be fig. 2D.

Authors' Response:

Corrected.

o Supplementary Figure 2b -The sequence is very difficult to read.

Authors' Response:

We have scaled up the sequence of VDR LBD in **Supplementary Fig.2b**.

Reviewer #3 (Remarks to the Author):

In an elegant application of hydrogen-deuterium exchange mass spectrometry, Zheng et al. demonstrate here how multiple signaling inputs (including DNA, ligand, and coregulator peptide) act combinatorially to selectively modulate structural dynamics and solvent accessibility of the VDRXR α heterodimer. They discriminate short- and long-range conformational perturbations conferred upon VDRXR α by the input signals individually and in combination. Framed by this foundational platform, the authors identify a selective VDR modulator (VDRM), Cmpd3; determine the VDR LBD-Cmpd3 structure by X-ray crystallography; model the VDR-LBD interaction with two other synthetic ligands; analyze by HDX MS the VDRXR α :Cmpd3 complex bound to two different VDR DNA binding sequences linked to 1,25D3-regulated genes important for bone formation and hypercalcemia; and show that these genes are regulated differentially by Cmpd3.

Against a matrix of VDR binding sites, ligands and a coregulator peptide, Zheng et al. compare VDRRXR α HDX-MS profiles under 28 distinct signaling conditions, demonstrating the conformational malleability of the receptor, with particular note to the extent and range of allosteric crosstalk. For example, binding to different DNA sequences alters HDX kinetics extending from the DBDs through the hinges and most extensively in the AF2 regions of the LBDs; similarly, specific ligand interactions trigger alterations in structural dynamics that extend into the DBDs, and changes in one heterodimer partner produce structural consequences in the other.

This comprehensive study will be broadly impactful for those invested in VDR mechanism and drug design, as well as transcriptional regulation, protein dynamics, structure, and allostery. In general, the work is clearly and logically presented, although there are several points that the authors should address prior to acceptance for publication:

Authors' Response:

We thank the reviewer for their positive comments on the manuscript.

Major points:

1. The authors extend a persistent failure in the field to define and distinguish properly the DNA and genomic components relevant to transcription and transcriptional regulation. These failures produce confusion, and more importantly misinterpretations and overinterpretations, producing misconceptions that inhibit progress. Corrections should be made throughout the manuscript, adopting the nomenclature and definitions below:

Vitamin D Response Element (VDRE): a genomic segment that confers a particular VDR response in vivo. In the present work, in vivo regulatory activity has not been validated for any of the investigated DNA segments, so VDRE is in no case appropriate terminology.

VDR Binding Sequence (VBS): a DNA sequence bound specifically, with high affinity, by VDR in vitro. A VBS has no ascribed function in vivo, either for VDR binding or for transcriptional regulation. The present work examines exclusively VBSs, none of which has been tested or validated as a VDRE; this should be corrected throughout.

VDR Occupied Region (VOR): a genomic segment occupied selectively by VDR in vivo. A VOR, typically detected as a "ChIP seq peak", carries no implication of transcriptional regulatory activity. In the present work, VORs are denoted as VDREs, as in "VDRRXR α translocates and associates with high binding affinity to genomic VDR response elements (VDREs)" (page 4). This should be corrected throughout; it would be appropriate to note that some of the VBSs studied are taken from VORs (although none of the VORs has been validated as a VDRE).

Promoter: a genomic segment at which general transcription factors, such as TFIIB, and RNA polymerase bind and initiate transcription of functional RNAs. Response elements, which bind transcriptional regulatory factors, such as VDR, rarely overlap even partially with promoters

(typically they are separated by 10-100kb or more), and the two entities are functionally distinct. In the present work, definitions of VDRE and promoter are conflated; this should be corrected throughout, including in the manuscript title.

General issues surrounding these definitions and functions are considered by Weikum et al. (Nature Rev Mol Cell Biol (2017) 18:159–74. Two authors of that perspective are responsible for this review, and would be happy to speak directly with Dr. Griffin and his coworkers if that would be useful in finalizing their manuscript.

Authors' Response:

We greatly appreciate the reviewer for pointing out the incorrect nomenclature used in our manuscript. We have noted and understand the definitions above, and we have made corrections throughout the text, figures, and legends to be consistent with the above nomenclature. In most cases, VDRE was changed to VBS.

2. Page 11: “The VDREs within the promoter region of *BGLAP* (5'-CTAGGTGAATGAGGACAT-3') and *TRPV6* (5'- CAAGGGGTAGTGAGGTCAAAAGCA-3') were chosen as they contain two direct repeats separated by a three base-pair gap, a signature highly reminiscent of canonical VDR response element 33,34.” This statement exemplifies some of the concerns noted above: the “signature” describes a family of closely related VBSs, i.e., a motif, not a “canonical VDR response element”; moreover, these are not part of “the promoter region”. The authors should state whether the VBSs tested are from genomic regions that are VDR occupied in vivo (i.e., are within VORs) in cells where the candidate target genes are VDR-responsive; the positions of selected VORs relative to candidate target genes should be provided. Notably, VBSs within the genome greatly outnumber VORs, which in turn outnumber VDREs; VORs differ in different cells, and VDREs can operate at very long range. Hence, proximity of VBSs or VORs to VDR-regulated genes identifies neither VDREs nor their target genes; this should be clarified/corrected throughout the manuscript. Validation of a VOR and resident VBS as a VDRE, and of a candidate target gene as a bona fide target gene, would require genome editing that produces allele-specific loss of regulation of the candidate target gene. It is of course interesting that VDRXR α binding to the selected VBSs provokes alterations in HDX kinetics. The reviewers accept that the authors may consider such validating CRISPR/Cas9 mutagenesis studies to be outside the scope of their already expansive investigation. If this is the case, they should state clearly that functional validation beyond VBS has not been done, and they should correct nomenclature accordingly.

Authors' Response:

We greatly appreciate the reviewer for pointing out the incorrect nomenclature used in our manuscript. As such we have edited all sections that previously used VDRE. See example below from Page 11.

“HDX was employed to probe the structural dynamics and activation mechanism of 1,25D3 and the non-calcemic modulator Cmpd3 liganded heterodimer bound to oligonucleotides representative of VBSs within the VDR target genes *BGLAP* and *TRPV6*. The sequences of the oligonucleotides used were derived from VBSs reported in *BGLAP* and *TRPV6*, each containing two direct repeats separated by a three base-pair gap, a motif of a family of closely related VBSs^{12,13}. For *BGLAP* two distinct oligos were used. The first was representative of a

VBS located at 457 b upstream from the transcription start site of *BGLAP* from rattus (5'-CTAGGTGAATGAGGACAT-3'), a VBS used in the previously reported SAXS study. A second oligo was representative of a VBS located at 510 b upstream of start site of human *BGLAP* gene (5'-GGTGA~~CT~~ACCGGGTGAACGGGGGCA-3')¹⁴. For *TRPV6*, an oligo representative of a VBS located 4.3 kb upstream from the transcription start site of human *TRPV6* gene (5'-CAAGGGGTAGTGAGGTCAAAGCA-3') was used. Binding of 1,25D3 liganded heterodimer to *TRPV6* VBS resulted in higher protection to solvent exchange in both the DBD and hinge domains of VDR as compared to that observed upon interaction with the *BGLAP* VBS, indicating that the *TRPV6* VBS makes more interactions with these domains than does the *BGLAP* VBS (**Fig. 5a and b** and **Supplementary Fig. 1b**, column (xvii) and (xviii)). VDR LBD exhibited only subtle perturbations in HDX behavior upon binding to the *BGLAP* VBS (**Fig. 5a and i** and **Supplementary Fig. 1b**, column (xvii)). This observation is consistent with the previous SAXS results that suggest the heterodimer complex forms an elongated conformation when bound to *BGLAP* VBS¹³. We then analyzed the protein complex bound to the human *BGLAP* VBS and obtained similar results to that with the rat *BGLAP* VBS (**Supplementary Fig. 1b**, column (xvii*)). Similarly, binding of 1,25D3 liganded heterodimer to *TRPV6* VBS displayed minor alterations in the dynamics of the VDR LBD (**Fig. 5b and i** and **Supplementary Fig. 1b** column (xviii)). Additionally, RXR α exhibited reduced deuterium exchange only in its DBD and dimer interface upon binding to *BGLAP* and *TRPV6* VBSs (**Fig. 5a and b** and **Supplementary Fig. 1b**, column (xvii) and (xviii)). Combined, these results further support that in the presence of the natural ligand 1,25D3, the heterodimer forms an open conformation upon binding to either *BGLAP* or *TRPV6* VBSs regardless of their specific nucleotide sequence."

In the Discussion we added the following text.

"It is important to note that the *BGLAP* and *TRPV6* VBSs used in these studies are putative VDR response elements (VDREs) yet to be validated as bona fide VDREs *in vivo*."

3. A BLAT search of each of the VBSs used in this study in the human genome (UCSC GRCh38/hg38) and BLAST through NCBI brought these results:

DR3 (5'-CGTAGGTCAATCAGGTCACGTCGT-3'): This sequence appears to be a synthetic construct composed of canonical binding motifs separated by a ATC spacer and flanked by 1-2 CTG repeats. It does not match any sequence in the human genome browser (UCSC genome browser human genome GRCh38/hg38). It should be explained as such. DR3 should not be italicized as it is throughout the text, as DR3 is not a gene name, but a synthetic DNA sequence "direct repeat with a 3 bp spacer".

Authors' Response:

We have removed the italic from DR3 and we modified its description as "a synthetic DNA sequence" in the text.

CYP24A1 (5'-CTAGCTCCCGAGGTCAGCGACGGCGCAGG-3'): No exact matches to this sequence were found within the human genome (UCSC) or in a BLAST is used on the Human Genomic + Transcript database (NCBI). Perhaps there is an altered region flanking a core VBS? Please explain.

Authors' Response:

We apologize for our incorrect identification of this oligo sequence. This oligo represents a single DR3 half-site and we changed the nomenclature to match this. This oligo is now referred to as “DR3 half-site”: a synthetic DNA sequence with half-site direct repeat. The purpose of this oligo was to see the impact on the heterodimer that one half-site would have as compared to DR3.

BGLAP (5'-CTAGGTGAATGAGGACAT-3'): This sequence is not found in UCSC genome browser human (GRCh38/hg38) or when BLAST is used on the Human Genomic + Transcript database. Please explain.

This was an oversight on our part that prompted us to perform additional experiments. The data from these new experiments has been included in the manuscript. Explanation: We used the *BGLAP* sequence (5'-CTAGGTGAATGAGGACAT-3') that was described and used in a SAXS study (an experimentally proven VBS)¹³. However, this sequence is from rat, not human *BGLAP*. The rat VBS *BGLAP* (5'-CTAGGTGAATGAGGACAT-3') is located at 457 b upstream from the transcription start site of rat *BGLAP*¹⁴. Given that our cell culture data is from a human cell line, we synthesized the human sequence and repeated all HDX studies that included the *BGLAP* VBS. The human VBS sequence we used for the new data is (5'-GGTGA CTACCGGGTGAACGGGGGCA-3') located at 510 b upstream of start site of human *BGLAP* gene¹⁴. Most importantly, the HDX data obtained on the VDRXR α heterodimer bound to either the rat or human *BGLAP* VBSs are very similar if not identical as described in the text (**Supplementary Figure 1a, b, c, d, e** column (xvii*), (xix*), (xxi*) and (xxiii*)). Thus, our conclusion that *BGLAP* VBS fails to induce long range allosteric communications to the VDR H12 region in both of 1,25D3 and VDRM bound VDRXR α complexes and results in an open H12 conformation, remains unchanged. The subsequent SRC1 RID binding resulted in similar protection to solvent exchange in H12 in both complexes, which is consistently different from that observed with *TRPV6* VBS binding.

Having the data on the rat *BGLAP* sequence is a bonus since the HDX data is consistent with the SAXS derived model with the heterodimer LBDs in an open conformation upon forming a complex with VBS *BGLAP* (5'-CTAGGTGAATGAGGACAT-3'). Hence it is useful to retain the rat data in our study.

TRPV6 (5'-CAAGGGGTAGTGAGGTCAAAGCA-3'): This sequence is located ~4.3 KB upstream from the transcription start site of *TRPV6* and underlies a VOR identified by ChIP. Meyer et al. (Mol Endo (2006) 20:1447–61) showed that this fragment, as well as numerous others underlying VORs proximal to *TRPV6*, confer VDR-mediated regulation in a reporter assay. Such plasmid-based reporter assays are not equivalent to genomic alterations introduced by Cas9-based editing, and are not sufficient to identify a functional VDRE. The authors may wish to denote them as “candidate VDREs”, although “VBSs” may be more accurate for this study.

Authors' Response:

We have corrected it as VBS *TRPV6* throughout the text.

4. The transcriptional data presented in Table 1 would be better displayed as a figure, with a

better explanation of how EC50 and %MaxStim were determined (along with measures of error in each experiment). This could be potentially being incorporated into Fig. 1.

Authors' Response:

In general, we agree; however, given the large amount of data obtained for we prefer to stay with the data in tabular form. To help explain the data we have recalculated the values to be normalized to the current assay data (1,25D3 is set to 100% for both BGLAP and TRPV6 within the data used in the manuscript) instead of being normalized to the historical values for 1,25D3 in these assays obtained many years earlier.

We added the following text to better explain the data in the table – “The maximum fold change over DMSO for cells treated with 1,25D3 was set to 100% maximum stimulation for both *BGLAP* and *TRPV6*, fold change for other compounds were normalized to the maximum stimulation value for 1,25D3 for each gene respectively.” As expected, the positive control for the assays, 1,25D3, was run many more times than the other compounds. Therefore, we present the average EC50 value obtained from all replicate runs (n = the number independent biological replicates).

The legend of Table 1 was modified as follows:

Chemical structures of ligands - natural agonist 1,25D3, Cmpd1, Cmpd2, and Cmpd3 - are shown with their respective EC50s for *BGLAP* and *TRPV6* gene activation assays, maximum stimulation value (normalized to the fold change of 1,25D3 treated cells over DMSO for both genes, and the number of independent biological replicates. ^aThe secondary alcohol stereocenter is a single unknown configuration. ^bBoth stereocenters are one single configuration. The configuration of the Ala stereocenter is known the secondary alcohol configuration is not. EC50 values are the average from the replicate runs.

Below is an example of representative data for compounds 2 and 3 in the MG63 *BGALP* qPCR assay.

	Cmpd3	Cmpd2
EC50	6.986e-007	3.478e-008

Ligand dependent activation of *BGLAP* mRNA in MG63 cells as determined by qPCR.

5. The HDX-MS “stabilization – destabilization” scale is shown only in the Supplement. As this is

central to all analyses, it should be included and explained briefly in Fig. 2 (and referenced in Figs 3-5).

Authors' Response:

We agree and we have updated relevant figures with HDX color key to indicate the stabilization and de-stabilization scale.

Minor points:

6. Figures contain ligands and DNA that are colored in hues that overlap with the HDX-MS scale, rendering it difficult to differentiate what is VDRXR α undergoing conformational changes and what is DNA or ligand (particularly for the orange ligand displayed in Figs. 2 and 5). Recoloring the ligands and DNA to be distinct from the HDX-MS range would facilitate interpretation of the results.

Authors' Response:

We agree and we have recolored the ligands and DNA to avoid a conflict with the colors used for illustrating the HDX behavior. Now 1,25D3 is colored purple and VDRM Cmpd3 is colored pink. DNA is colored black throughout the figure.

7. In the Supplement, the authors consider different VDRXR α models derived from X-ray crystallography and from cryo-EM/SAXS. It would be helpful to mention more prominently these models in the main text, and to discuss the “fit” of each to the HDX-MS data.

Authors' Response:

We agree and this prompted us to regenerate the VDRXR α model using the cryo-EM structural model.

The following text was added/modified to describe how the model was made:

“A structural model of the VDR (amino acids 18-455)-RXR (amino acids 135-462) heterodimer bound to the DR3 DNA fragment was created using Molsfot ICM Pro software. Amino Acids 201-223 of RXR and 166-216 of VDR could not be modelled due lack of available homology models containing these amino acid residues. The heterodimer was created in the “open” conformation and a partial cryo-EM model (generous gift from Professor Dino Moras) was used to guide the creation of the model. Loops that could not be modelled from the cryo-EM data were added (mostly in the hinge regions) and hydrogen atoms were added to all amino acids and both ligands. The added loops were subject to loop modelling to improve clashing using the ICM-Pro loop modelling utility. The final structure was subjected to minimization, regularization, and annealing in ICM Pro.”

The model was added to **Supplementary Figure 2c** along with the legend.

Legend: VDR/RXRα DR3 model: the VDR/RXRα heterodimer was created in the “open” conformation and a partial cryo-EM model (generous gift from Professor Dino Moras) was used to guide the creation of the model. VDR/RXRα heterodimer forms an extended, L-shaped organization with RXRα DBD occupying one half-site of DR3 at 5'- end and VDR DBD occupying another at 3'- end^{5,6}. VDR hinge domain, as a connection linker between DBD and LBD, plays a key role in determining structural orientation of LBD dimer, which is arranged perpendicular to DR3. Residue Proline122, which resides on the VDR hinge domain, was described as the “kink residue” between the C-terminal hinge helix and helix H1 of the LBD playing an important role in dictating the orientation of the LBD, resulting in an open LBD dimer architecture facing away from DBD. In this model, VDR hinge domain adopts a α -helical structure residing closely to DR3 and makes extensive contacts with phosphate and backbone of DNA⁶. This is in consistency with our HDX observations that VDR hinge undergoes different extent of protections upon binding to various VBSs. Unlike VDR, DNA binding does not perturb RXRα hinge dynamics as it forms a flexible linker that allows enhanced adaptability of heterodimer to diverse response elements. This model suggests that the hinges could possess intrinsic properties to orient LBD dimer in a precise way and provide structural basis for crosstalk between LBDs and DBDs.

8. It would be helpful to expand the explanation of EX1 and EX2 kinetics beyond “occurrence of EX1 kinetics in protein native state has been shown to provide important clues to protein intermediate conformational states in solution^{22,39,40}”.

Authors' Response:

We agree and we have added the following text to expand the discussion of EX1 and EX2 kinetics in the main text.

“For EX1 kinetics (cooperative unfolding) the refolding rate of the corresponding protein region is slower than the solvent exchange rate of amide hydrogens in the unfolded region, thus these

amide hydrogens exchange simultaneously. This situation gives rise to a distinct MS signature, specifically bimodal mass distributions, with the lower mass envelop corresponding to molecules that have not yet exchanged (molecules that have not yet unfolded) and the higher mass envelop corresponding to molecules that have undergone solvent exchange (molecules that have unfolded). Under native state conditions, most regions of proteins exhibit EX2 kinetics wherein only a single MS envelop is observed over time (the refolding rate is faster than the rate of solvent exchange). However, the occurrence of EX1 cooperative unfolding behavior in native state proteins have been shown to provide important clues to intermediate conformational states of proteins in solution.”

9. The rationale for displaying some model structures in Figs. 2 - 5 in ribbon and some in surface is presumably to differentiate SRC1-bound complexes from those lacking SRC1. In any case, some explanation should be added to the legends. In addition, the surface renderings would benefit from an outline of the boundaries of each protein, so readers could more readily discern which receptor surfaces display altered HDX-MS dynamics.

Authors' Response:

We agree and we have remade the VDRXR α homology model as described above and fixed the figure legend describing whether the relevant model is shown in ribbon or in surface. The color used in the respective model shown in Figure 1-Figure 5 are mapped from HDX perturbation view it is difficult to change the color scheme to clearly show each protein boundary so in the updated figure we added arrows to mark which receptor the AF2 protection is observed.

10. Simple edits:

Capital C in Cmpd throughout.

Authors' Response:

Corrected.

Fig. 1d legend asserts that “VDR LBD is shown in ribbons” whereas ribbon is not shown and side chains are displayed in ball and stick.

Figs. 2-5, each legend: “Percentages of deuterium differences are color-coded according to the smooth color gradient key. Dark gray, no statistically significant changes between compared conditions; light gray, regions that have no sequence coverage and include proline residue that has no amide hydrogen exchange activity.” Include this statement in Fig. 2 legend only; for Figs. 3-5, use “Percentages of deuterium differences are coded as in Fig 2.”

Authors' Response:

Corrected.

Ensure that Methods section is consistent in terminology and nomenclature. In the present version, for example, the Reagents subsection cites Ni-NTASEC and denotes a deletion as “ Δ ”, whereas the Crystallography subsection refers to “Nickel chelate affinity chromatography and

size-exclusion chromatography”, and denotes a deletion as “del.”

Authors' Response:

Noted. Have unified as Ni-NTASEC and Δ.

References used in this response letter:

1. Fleet, J.C. & Schoch, R.D. Molecular mechanisms for regulation of intestinal calcium absorption by vitamin D and other factors. *Crit Rev Clin Lab Sci* **47**, 181-95 (2010).
2. Christakos, S., Dhawan, P., Verstuyf, A., Verlinden, L. & Carmeliet, G. Vitamin D: Metabolism, Molecular Mechanism of Action, and Pleiotropic Effects. *Physiol Rev* **96**, 365-408 (2016).
3. Banerjee, P. & Chatterjee, M. Antiproliferative role of vitamin D and its analogs--a brief overview. *Mol Cell Biochem* **253**, 247-54 (2003).
4. van de Peppel, J. & van Leeuwen, J.P. Vitamin D and gene networks in human osteoblasts. *Front Physiol* **5**, 137 (2014).
5. Nwachukwu, J.C. & Nettles, K.W. The nuclear receptor signalling scaffold: insights from full-length structures. *EMBO J* **31**, 251-3 (2012).
6. Orlov, I., Rochel, N., Moras, D. & Klaholz, B.P. Structure of the full human RXR/VDR nuclear receptor heterodimer complex with its DR3 target DNA. *EMBO J* **31**, 291-300 (2012).
7. Weis, D.D., Wales, T.E., Engen, J.R., Hotchko, M. & Ten Eyck, L.F. Identification and characterization of EX1 kinetics in H/D exchange mass spectrometry by peak width analysis. *J Am Soc Mass Spectrom* **17**, 1498-509 (2006).
8. Guttman, M., Weis, D.D., Engen, J.R. & Lee, K.K. Analysis of overlapped and noisy hydrogen/deuterium exchange mass spectra. *J Am Soc Mass Spectrom* **24**, 1906-12 (2013).
9. Heery, D.M., Kalkhoven, E., Hoare, S. & Parker, M.G. A signature motif in transcriptional co-activators mediates binding to nuclear receptors. *Nature* **387**, 733-6 (1997).
10. Cummins, D.J. et al. Two-Site Evaluation of the Repeatability and Precision of an Automated Dual-Column Hydrogen/Deuterium Exchange Mass Spectrometry Platform. *Anal Chem* **88**, 6607-14 (2016).
11. Pascal, B.D. et al. HDX workbench: software for the analysis of H/D exchange MS data. *J Am Soc Mass Spectrom* **23**, 1512-21 (2012).
12. Meyer, M.B., Watanuki, M., Kim, S., Shevde, N.K. & Pike, J.W. The human transient receptor potential vanilloid type 6 distal promoter contains multiple vitamin D receptor binding sites that mediate activation by 1,25-dihydroxyvitamin D3 in intestinal cells. *Mol Endocrinol* **20**, 1447-61 (2006).
13. Rochel, N. et al. Common architecture of nuclear receptor heterodimers on DNA direct repeat elements with different spacings. *Nat Struct Mol Biol* **18**, 564-70 (2011).
14. Ozono, K., Liao, J., Kerner, S.A., Scott, R.A. & Pike, J.W. The vitamin D-responsive element in the human osteocalcin gene. Association with a nuclear proto-oncogene enhancer. *J Biol Chem* **265**, 21881-8 (1990).

REVIEWERS' COMMENTS:

Reviewer #1 (Remarks to the Author):

Authors carefully addressed the comments raised in my first review. Therefore, I recommend this manuscript for publication in Nature Communication.

Reviewer #2 (Remarks to the Author):

In the revised version of the manuscript, the additional supporting information and the detailed cover letter, the authors fully and adequately address my comments. I have no further requests and recommend publication.

Reviewer #3 (Remarks to the Author):

The authors have thoroughly and conscientiously responded to our comments and concerns. We congratulate them on their beautiful and important work.

A few minor points to be considered during finalization for publication:

1. Line 64: "upon binding specific regions within promoters of VDR target genes." Replace "within promoters", perhaps with "thought to regulate"
2. Lines 343-344: "VBSs within the target genes BGLAP and TRPV6." Replace perhaps with "VBSs linked to the BGLAP and TRPV6 genes."
3. Lines 384 and 400: The nomenclature for helix 12 should be normalized throughout the text, e.g., helix12 is used in line 384, while H12 used in line 400.
4. Line 668 "... smooth color gradient key (Supplementary Fig.2b)" The color key is now placed at the bottom of Fig. 1 and should be referenced here. Also, a unit label on the color key would be helpful, such as "% deuterium differences" within the figure. For consistency the same color key and statements could be made in Figs 1-5, referring to Fig. 1 as the primary explanation. In the current draft, Figs. 3-5 refer to key in Fig 2.
5. Fig. 4 – Please add color key as in Fig.1
6. Fig.5 (line 730) "(shown in ribbon), and SRC1 RID (shown in surface) (on the right)" would be more clear if changed to "(shown in ribbon a-d), and SCR1 RID (shown in surface e-h) (on the right)"

Submitted by Keith Yamamoto and Matthew Kneusel